# The transition to synchronization of networked systems

Atiyeh Bayani [1], Fahimeh Nazarimehr [1], Sajad Jafari [1,2] ✉, Kirill Kovalenko [3], Gonzalo Contreras-Aso [4], Karin Alfaro-Bittner [4] ✉, Rubén J. Sánchez-García [5,6,7] ✉ & Stefano Boccaletti [8,9,10]

We study the synchronization properties of a generic networked dynamical system, and show that, under a suitable approximation, the transition to synchronization can be predicted with the only help of eigenvalues and eigenvectors of the graph Laplacian matrix. The transition comes out to be made of a well defined sequence of events, each of which corresponds to a specific clustered state. The network's nodes involved in each of the clusters can be identified, and the value of the coupling strength at which the events are taking place can be approximately ascertained. Finally, we present large-scale simulations which show the accuracy of the approximation made, and of our predictions in describing the synchronization transition of both synthetic and real-world large size networks, and we even report that the observed sequence of clusters is preserved in heterogeneous networks made of slightly non-identical systems.

From brain dynamics and neuronal firing, to power grids or financial markets, synchronization of networked units is the collective behavior characterizing the normal functioning of most natural and man-made systems[1–7]. As a control parameter (typically the coupling strength in each link of the network) increases, a transition occurs between a fully disordered and gaseous-like phase (where the units evolve in a totally incoherent manner) to an ordered or solid-like phase (in which, instead, all units follow the same trajectory in time).

The transition between such two phases can be discontinuous and irreversible, or smooth, continuous, and reversible. The first case is known as Explosive Synchronization[8], which has been described in various circumstances[9–13], and which refers to an abrupt onset of synchronization following an infinitesimally small change in the control parameter, with hysteresis loops that may be observed as in a thermodynamic first-order phase transition. The second case is the most commonly observed one, and corresponds instead to a second-order

phase transition, resulting in intermediate states emerging in between the two phases. Namely, the path to synchrony[14] is here characterized by a sequence of events where structured states emerge made of different functional modules (or clusters), each one evolving in unison. This is known as cluster synchronization (CS)[15–17], and a lot of studies pointed out that the structural properties of the graph are responsible for the way nodes clusterize during CS[18–21]. In particular, it was argued that the clusters formed during the transition are to be connected to the symmetry orbits and/or to the equitable partitions of the graph[19,22]. Precisely, partitions of the network's nodes into orbits (for the whole, or a subgroup, of the automorphism group of the graph) or, more generally, equitable partitions, have been identified as potential clusters that can synchronize before the setting of complete synchronization in the network[23,24]. The fact that a given graph partition can sustain cluster synchronization of its cells is different, however, from the problem faced in our study i.e., that of determining whether it will

[1]Department of Biomedical Engineering, Amirkabir University of Technology (Tehran Polytechnic), Tehran, Iran. [2]Health Technology Research Institute, Amirkabir University of Technology (Tehran Polytechnic), Tehran, Iran. [3]Scuola Superiore Meridionale, School for Advanced Studies, Naples, Italy. [4]Universidad Rey Juan Carlos, Calle Tulipán s/n, Madrid, Spain. [5]Mathematical Sciences, University of Southampton, Southampton, UK. [6]Institute for Life Sciences, University of Southampton, Southampton, UK. [7]The Alan Turing Institute, London, UK. [8]CNR - Institute of Complex Systems, Sesto Fiorentino, Italy. [9]Sino-Europe Complexity Science Center, School of Mathematics, North University of China, Shanxi, Taiyuan, China. [10]Research Institute of Interdisciplinary Intelligent Science, Ningbo University of Technology, Zhejiang, Ningbo, China. ✉e-mail: sajadjafari83@gmail.com; karin.alfaro@urjc.es; rsanchez-garcia@turing.ac.uk

indeed be realized for some range of the coupling strength parameter and, more importantly, in which order do clusters synchronize during the transition between the two main phases (i.e., as the coupling strength increases from zero).

On the other hand, clusters due to symmetries are found rather ubiquitously, as most of real-world networks have a large number of localized symmetries, that is, a large number of small subgraphs called symmetric motifs[25,26], where independent (formally, support-disjoint) symmetries are generated, with each symmetric motif made of one or more orbits. For instance, ref. 26 analyzes several real-world networks and, in all cases, gives evidence of a large number of symmetric motifs: from 149 motifs in the protein-protein interaction network of the yeast (a network of 1647 nodes) to over 245,000 motifs in the LiveJournal social network (more than 5 million nodes). More evidence of symmetries in real-world networks can be found in refs. 27–29.

In our work we assume that, during the transition, the synchronous solution of each cluster does not differ substantially from that of the entire network and, under such an approximation, we elaborate a practical technique which is able to elucidate the transition to synchronization in a generic network of identical systems. Namely, we introduce a (simple, effective, and limited in computational demand) method which is able to: (i) predict the entire sequence of events that are taking place during the transition, (ii) identify exactly which graph's node is belonging to each of the emergent clusters, and (iii) provide a well approximated calculation of the critical coupling strength value at which each of such clusters is observed to synchronize. We also demonstrate that, under the assumed approximation, the sequence of events is in fact universal, in that it is independent on the specific dynamical system operating in each network's node and depends, instead, only on the graph's structure. Our study, moreover, allows to clarify that the emerging clusters are those groups of nodes which are indistinguishable at the eyes of any other network's vertex. This means that all nodes in a cluster have the same connections (and the same weights) with nodes not belonging to the cluster, and therefore they receive the same dynamical input from the rest of the network. As such, we prove that synchronizable clusters in a network are subsets more general than those defined by the graph's symmetry orbits, and at the same time more specific than those described by equitable partitions (see the Supplementary Information for a detailed description of the exact relationship between the observed clusters and the graph's symmetry orbits and equitable partitions). Finally, we present extensive numerical simulations with both synthetic and real-world networks, which demonstrate the high accuracy of our approximations and predictions, and also report on synchronization features in heterogeneous networks showing that the predicted cluster sequence can be maintained even for networks made of non-identical dynamical units.

## Results

### The synchronization solution

The starting point is a generic ensemble of $N$ identical dynamical systems interplaying over a network $G$. The equations of motion are

$$\dot{\mathbf{x}}_i = \mathbf{f}(\mathbf{x}_i) - d\sum_{j=1}^{N} \mathcal{L}_{ij}\,\mathbf{g}(\mathbf{x}_j), \tag{1}$$

where $\mathbf{x}_i(t)$ is the $m$-dimensional vector state describing the dynamics of each node $i$, $\mathbf{f} : \mathbb{R}^m \longrightarrow \mathbb{R}^m$ describes the local (identical in all units) dynamical flow, $d$ is a real-valued coupling strength, $\mathcal{L}_{ij}$ is the $(ij)$ entry of the Laplacian matrix associated to $G$, and $\mathbf{g} : \mathbb{R}^m \longrightarrow \mathbb{R}^m$ is the output function through which units interact. $\mathcal{L}$ is a zero-row matrix, a property which, in turn, guarantees existence and invariance of the network's synchronized solution $\mathbf{x}_s(t) = \mathbf{x}_1(t) = \ldots = \mathbf{x}_N(t)$.

The necessary condition for the stability of such solution can be assessed by means of the Master Stability Function (MSF) approach, a

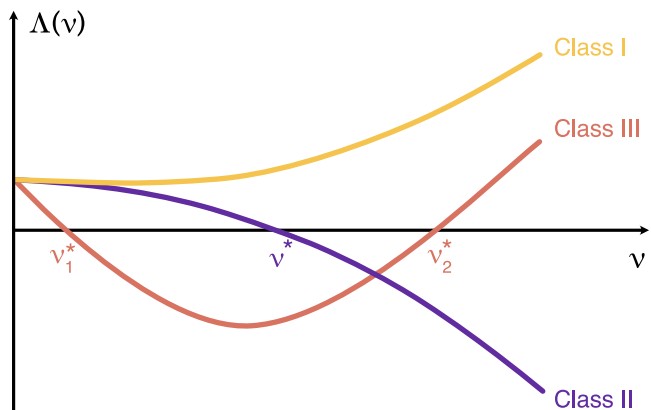

**Fig. 1 | Classes in the Master Stability Function.** The maximum Lyapunov exponent $\Lambda$ as a function of the parameter $\nu$ (see text for definitions), for a chaotic flow ($\Lambda(0) > 0$). The curve $\Lambda(\nu)$ is called the Master Stability Function (MSF). Given any pair of $\mathbf{f}$ and $\mathbf{g}$, only three classes of systems are possible: Class I systems (yellow line) for which the MSF does not intercept the horizontal axis; Class II systems (violet line), for which the MSF has a unique intercept with the horizontal axis at $\nu = \nu^*$; Class III systems (brown line), for which the MSF intercepts the horizontal axis at two critical points $\nu = \nu_1^*$ and $\nu = \nu_2^*$.

method initially developed for pairwise coupled systems[30], and later extended in many ways to heterogeneous networks[31], and to time-varying[32] and higher-order[33] interactions. When possible, the MSF formalism can be complemented by other global approaches (such as the Lyapunov functionals) which provide sufficient conditions for stability. As the MSF is of rather standard use, all details about the associated mathematics are contained in the Methods section. Notice, moreover, that our Eq. (1) is in fact written in the simplest as possible form for the easiness of the mathematical passages that will be described below. However, our approach can be extended straightforwardly (see details in refs. 33,34 and references therein) to the much more general case of a coupling term given by $d\sum_{j=1}^{N} \mathcal{A}_{ij}\,\mathbf{g}(\mathbf{x}_i, \mathbf{x}_j)$, where $\mathcal{A}_{ij}$ is the $(ij)$ entry of the graph's adjacency matrix, under only two assumptions: (i) the Laplacian matrix $\mathcal{L}$ associated to $\mathcal{A}$ is diagonalizable, and the set of eigenvectors form an orthonormal basis of $\mathbb{R}^N$, and (ii) the coupling function $\mathbf{g}(\mathbf{x}_i, \mathbf{x}_j)$ is synchronization non-invasive i.e., it strictly vanishes at the synchronization solution ($\mathbf{g}(\mathbf{x}_s, \mathbf{x}_s) = 0$).

The full details behind the MSF approach can be found in the Methods section. We here concisely summarize them to pave the way for the cluster synchronization analysis which will follow. In essence, one considers perturbations around the synchronous state, and performs linear stability analysis of Eq. (1). Due to the properties of $\mathcal{L}$, the perturbations can be expanded as linear combinations of its eigenvectors $\mathbf{v}_i$ (which span the space $\mathcal{T}$ tangent to the synchronization manifold $\mathcal{M}$). The coefficients $\eta_i$ of this expansion obey variational equations involving the Jacobians of $\mathbf{f}$ and $\mathbf{g}$, which only differ to one another for the value of the corresponding eigenvalue $\lambda_i$. This fact allows one to parameterize the problem, and to refer to a unique variational equation which is now an explicit function of the parameter $\nu = d\lambda_i$. The maximum Lyapunov exponent $\Lambda$ of such a parametric equation can then be computed for increasing values of $\nu$. The function $\Lambda(\nu)$ is the Master Stability Function (MSF), and its values assess the expansion (if positive) or contraction (if negative) rates in the directions of eigenvectors $\mathbf{v}_i$. One needs all of these rates to be negative in order for the synchronization manifold $\mathcal{M}$ to be stable.

As noticed for the first time in Chapter 5 of ref. 4, and as illustrated in Fig. 1, all possible choices of (chaotic) flows and output functions, are in fact categorizable into only three classes of systems:

- Class I systems, for which the MSF does not intercept the horizontal axis, like the yellow curve in Fig. 1. These systems

intrinsically defy synchronization, because all directions of $\mathcal{T}$ are always (i.e., at all values of $d$) expanding, no matter which network architecture is used for connecting the nodes. Therefore, neither the synchronized solution $\mathbf{x}_s(t)$ nor any other cluster-synchronized state will ever be stable.

- Class II systems, for which the MSF has a unique intercept with the horizontal axis at a critical value $v = v^*$, like the violet line of Fig. 1. The scenario here is the opposite of that of Class I. Indeed, given any network $G$, the condition $d\lambda_2 > v^*$ warrants stability of the synchronized solution. These systems, therefore, are always synchronizable, and the threshold for synchronization is $d_c \equiv \frac{v^*}{\lambda_2}$ i.e., is inversely proportional to the second smallest eigenvalue of the Laplacian matrix.
- Class III systems, for which the MSF intercepts instead the horizontal axis at two critical points $v = v_1^*$ and $v = v_2^*$, like the brown V-shaped curve of Fig. 1. In order for the synchronization solution to be stable, it is required in this case that the entire spectrum of eigenvalues of $\mathcal{L}$ falls (when multiplied by $d$) in between $v_1^*$ and $v_2^*$. In other words, the two conditions $d\lambda_N < v_2^*$ and $d\lambda_2 > v_1^*$ must be simultaneously verified, and this implies that not all networks succeed to synchronize Class III systems. In fact, the former condition gives a bound $d_{max} = \frac{v_2^*}{\lambda_N}$ for the coupling strength above which instabilities in tangent space start to occur in the direction of $\mathbf{v}_N$, the latter provides once again the threshold $d_c \equiv \frac{v_1^*}{\lambda_2}$ for complete synchronization to occur.

In general, one should point out that the region of stability could even be formed by the union of several intervals on the $v$ axis. However, the aim of present work is to describe the first transition to synchronization (i.e., the process that starts at $d = 0$ and occurs when progressively increasing the coupling strength) for which the three mentioned classes are indeed the only possible scenarios. When a system will show multiple regions of stability in the $v$ axis, instead, a series of synchronization and de-synchronization transitions (as many as the stability regions of the system) will occur, and this latter scenario will be reported by us elsewhere.

Finally, it is crucial to remark that all the above results are formally valid only for the whole network's synchronous solution. The trajectories followed by the nodes' dynamics in each cluster-synchronous state slightly differ, instead, from those which are followed in the global solution, as they rigorously depend on the quotient network (and therefore on topology, node dynamics, and clusterization, see the Methods section for a detailed description of such differences), and several ad-hoc methods have been proposed to assess the stability of synchronization in each specific clustered state[19,22,35–38].

However, since we are focusing on the transition to synchronization (i.e., on a regime where the coupling strength is normally very small), in the following we will adopt the approximation that such a difference will only be tight, and will not give rise to substantial changes in the calculation of maximum Lyapunov exponents. As a consequence, we will refer to the MSF calculated for the full network's synchronous solution as a an approximation of the values of the maximum Lyapunov exponents corresponding to the eigenvectors orthogonal to each cluster's synchronous solutions. Notice, furthermore, that at each value of the coupling strength one might have a distinct cluster-synchronous state.

## The path to synchronization
With all this in mind, let us now move to describe all salient features characterizing the transition to the synchronization solution (as $d$ increases from 0), and in particular to predict all the intermediate events that are taking place during such a transition. Since now, we anticipate that our results are valid for all systems in Class II, as well as

for those in Class III (up to the maximum allowed value of the coupling strength i.e., for $d < \frac{v_2^*}{\lambda_N}$).

There are three conceptual steps that need to be made.

The first step is that, as $d$ progressively increases, the eigenvalues $\lambda_i$ cross the critical point ($v = v^*$ in Class II, or $v = v_1^*$ in Class III) sequentially. The first condition which will be met will be, indeed, $d\lambda_N > v^*$ in Class II ($d\lambda_N > v_1^*$ in Class III), while for larger values of $d$ the other eigenvalues will cross the critical point one by one (if they are not degenerate) and in the reverse order of their size.

Therefore, one can use this very same order to progressively unfold the tangent space $\mathcal{T}$. In particular, at any value of $d$, $\mathcal{T}$ can be factorized as $\mathcal{T}^+(d) \otimes \mathcal{T}^-(d)$, where $\mathcal{T}^+(d)$ [$\mathcal{T}^-(d)$] is the subspace generated by the set of eigenvectors $\{\mathbf{v}_i\}$ whose corresponding $d\lambda_i$ are below (above) the stability condition, i.e. for which one has $d\lambda_i \leq v^*$ ($d\lambda_i > v^*$) in Class II, or $d\lambda_i \leq v_1^*$ ($d\lambda_i > v_1^*$) in Class III. In other words, the subspace $\mathcal{T}^+(d)$ [$\mathcal{T}^-(d)$] contains only expanding (contracting) directions, and therefore the projection on it of the synchronization error $\delta\mathbf{X}$ will exponentially increase (shrink) in size.

The second step consists in taking note that, if one constructs the matrix $V$ having as columns the eigenvectors $\mathbf{v}_i = (v_{i1}, v_{i2}, \ldots, v_{iN})$, that is

$$
V = \begin{pmatrix}
v_{11} & v_{21} & \cdots & v_{N1} \\
v_{12} & v_{22} & \cdots & v_{N2} \\
\vdots & \vdots & \ddots & \vdots \\
v_{1N} & v_{2N} & \cdots & v_{NN}
\end{pmatrix},
\tag{2}
$$

then the rows of matrix (2) provide an orthonormal basis of $\mathbb{R}^N$ as well, since $V^T V = \mathbb{I}$ implies $V^T = V^{-1}$, and hence also $VV^T = \mathbb{I}$.

Therefore, one can now examine the eigenvectors component-wise and, for each eigenvector $\mathbf{v}_i$, define the following matrix

$$
E_{\lambda_i} = \begin{pmatrix}
(v_{i1} - v_{i1})^2 & (v_{i2} - v_{i1})^2 & \cdots & (v_{iN} - v_{i1})^2 \\
(v_{i1} - v_{i2})^2 & (v_{i2} - v_{i2})^2 & \cdots & (v_{iN} - v_{i2})^2 \\
\vdots & \vdots & \ddots & \vdots \\
(v_{i1} - v_{iN})^2 & (v_{i2} - v_{iN})^2 & \cdots & (v_{iN} - v_{iN})^2
\end{pmatrix}.
$$

Furthermore, following the same sequence which is progressively unfolding $\mathcal{T}$, one recursively defines the following set of matrices $S_n$

$$
\begin{aligned}
S_N &= E_{\lambda_N}, \\
S_{N-1} &= S_N + E_{\lambda_{N-1}}, \\
&\ldots, \\
S_2 &= S_3 + E_{\lambda_2}, \\
S_1 &= S_2 + E_{\lambda_1}.
\end{aligned}
$$

It is worth discussing a few properties of such matrices. First of all, as $\mathbf{v}_1$ is aligned with $\mathcal{M}$, all its components are equal, and therefore $E_{\lambda_1} = 0$ and $S_1 = S_2$. Second, all the $E$-matrices, and thus all the $S$-matrices, are symmetric, non negative and have all diagonal entries equal to zero. In fact, the off diagonal $ij$ elements of the matrix $S_n$ ($n = 1, \ldots, N$) are nothing but the square of the norm of the vector obtained as the difference between the two vectors defined by rows $i$ and $j$ of matrix (2), truncated to their $n$ last components. As so, the maximum value that any entry ($ij$) may have in the matrices $S_n$ is 2, which corresponds to the case in which such two vectors are orthonormal. In particular, all off-diagonal entries of $S_2 = S_1$ are equal to 2.

The third conceptual step consists in considering the fact that the Laplacian matrix $\mathcal{L}$ uniquely defines $G$, and as so any clustering property of the network $G$ has to be reflected into a corresponding spectral feature of $\mathcal{L}$[26,27]. In this paper, we can prove rigorously that the synchronized clusters emerging during the transition of $G$ can be

associated to a localization of a group of the Laplacian eigenvectors on the clustered nodes. Namely, let us first define that a subset $\mathcal{S} \subseteq \{\mathbf{v}_2, \mathbf{v}_3, \ldots, \mathbf{v}_N\}$ consisting of $k-1$ eigenvectors forms a spectral block localized at nodes $\{i_1, \ldots, i_k\}$ if

- each eigenvector belonging to $\mathcal{S}$ has all entries (except $i_1, i_2, \ldots, i_k$) equal to 0;
- for each other eigenvector $\mathbf{v}$ not belonging to $\mathcal{S}$, the entries $i_1, i_2, \ldots, i_k$ are all equal i.e., $v_{i_1} = v_{i_2} = \ldots = v_{i_k}$.

Moreover, all eigenvectors $\{\mathbf{v}_2, \mathbf{v}_3, \ldots, \mathbf{v}_N\}$ are orthonormal to $\mathbf{v}_1$, and therefore the sum of all their entries must be equal to 0.

The main theoretical result underpinning our study is the Theorem stated below.

**Theorem.** The following two statements are equivalent:

1. All $k$ nodes belonging to a cluster defined by the indices $\{i_1, \ldots, i_k\}$ have the same connections with the same weights with all other nodes not belonging to the cluster, i.e. for any $p, q \in \{i_1, \ldots, i_k\}$ and $j \notin \{i_1, \ldots, i_k\}$ one has $\mathcal{L}_{pj} = \mathcal{L}_{qj}$.
2. There is a spectral block $\mathcal{S}$ made of $k-1$ Laplacian's eigenvectors localized at nodes $\{i_1, \ldots, i_k\}$.

A group of nodes satisfying condition (1) of the theorem is also called an external equitable cell[39].

The reader interested in the mathematical proof of the theorem is referred to our Supplementary Information. We here concentrate, instead, on the main concepts involved. Conceptually, the first statement of the Theorem is tantamount to assert that the nodes belonging to a given cluster are indistinguishable to the eyes of any other node of the network, but puts no constraints on the way such nodes are connected among them within the cluster. Therefore, fulfillment of the statement is realized by (but not limited to) the case of a network's symmetry orbit. In other words, the first statement of the theorem says that the clustered nodes receive an equal input from the rest of the network, and therefore (for the principle that a same input will eventually - i.e. at sufficiently large coupling -imply a same output) they may synchronize independently on the synchronization properties of the rest of the graph. Therefore, the intermediate structured states emerging in the path to synchrony of a network are more general than the graph's symmetry orbits, but more specific than the graph's equitable partitions.

However, the most relevant consequence of the theorem is that the localization of the eigenvectors' components implies that the matrices $S_n$ may actually display entries equal to 2 also for $n$ strictly larger than 2! Indeed, the $(ij)$ entry of the matrices $S_n$ is just equal to

$$\sum_{k=n}^{N} (v_{kj} - v_{ki})^2.$$

Now, suppose that $\mathcal{S}$ is a spectral block localized at $i, j$ and some other nodes. Then, if $\mathbf{v}_k$ does not belong to $\mathcal{S}$, the term $(v_{kj} - v_{ki})^2 = 0$, and one has therefore that the $(ij)$ entry of $S_n$ has contributions only from those eigenvectors $\mathbf{v}_n$ belonging to $\mathcal{S}$. Now, all the times that a localized spectral block $\mathcal{S}$ is contained in the set of those eigenvectors generating $\mathcal{T}^-(d)$ the corresponding cluster of nodes will emerge as stable synchronization cluster, because the tangent space of the corresponding synchronized solution (where synchrony is limited to those specific nodes) can be fully disentangled from the rest of $\mathcal{T}$ and will consist, moreover, of only contractive directions.

**Complete description of the transition**

By exploiting the outcomes of the three conceptual steps described above, an extremely simple (and computationally low demanding) technique can then be introduced, able to monitor and track localization of eigenvectors along the transition, and therefore to describe the path to synchronization.

The method consists in the following steps:

- given a network $G$, one considers the Laplacian matrix $\mathcal{L}$, and extracts its $N$ eigenvalues $\lambda_i$ (ordered in size) and the corresponding eigenvectors $\{\mathbf{v}_i\}$;
- one then calculates the $N$ matrices $E_{\lambda_i}$ and $S_n$ ($i = 1, \ldots, N; n = 1, \ldots, N$);
- one inspects the matrices $S_n$ in the same order with which the Laplacian's eigenvalues (when multiplied by $d$) crosses the critical point $v^*$ (i.e., $N, N-1, N-2, \ldots, 2, 1$), and looks for entries which are equal to 2;
- when, for the first time in the sequence (say, for index $p$) an entry in matrix $S_p$ is (or multiple entries are) found equal to 2, a prediction is made that an event will occur in the transition: the cluster (or clusters) formed by the nodes with labels equal to those of the found entry (entries) will synchronize at the coupling strength value $v^*/\lambda_p$. The inspection of matrices $S_n$ then continues, focusing only on the entries different from those already found to be 2 at level $S_p$;
- after having inspected all $S_n$ matrices, one obtains the complete description of the sequence of events occurring in the transition, with the exact indication of all the values of the critical coupling strengths at which each of such events is occurring. By events, we here mean either the formation of one (or many) new synchronized cluster(s), or the merging of different clusters into a single synchronized one.

Once again, we here remark that our results and methods are valid under the approximation that all cluster-synchronous solutions are well described by the synchronization state that each node in the cluster would display in the complete synchronization scenario i.e., when the entire network synchronizes. In the following, we will demonstrate the accuracy of such an approximation with respect to synthetic and real-world networks. We begin with illustrating the method in a simple case, in order for the reader to have an immediate understanding of the consequences of the various steps that have been discussed so far. For this purpose, we consider the network sketched in panel (a) of Fig. 2, which consists of an all-to-all connected, symmetric, weighted graph of $N = 10$ nodes (see the Supplementary Information for the adjacency matrix of the network). By construction, the graph is endowed with three symmetric orbits: the first being composed by the pink nodes $1, 2$ and $3$, the second containing the blue nodes $4, 5$ and $6$, and the third being made of the four green nodes $7, 8, 9$ and $10$. The 10 eigenvalues of the Laplacian matrix, when ordered in size, are $\{0, 1, 1, 1, 4, 4, 4, 6, 6, 6\}$.

After calculations of the corresponding eigenvectors, the matrices $E_{\lambda_i}$ and $S_n$ are evaluated. Then, one starts inspecting matrices $S_n$ in the reverse order of the size of the corresponding eigenvalues. Figure 2b shows $S_{10}$, which corresponds to $\lambda_{10} = 6$, and it can be seen that there are no entries equal to 2 in such a matrix. Nor entries equal to 2 are found in $S_9$ (not shown). However, when inspecting $S_8$ (which corresponds to $\lambda_8 = 6$), one immediately identifies [Fig. 2c] many entries equal to 2, which clearly define a cluster formed by nodes $7, 8, 9$ and $10$. A prediction is then made that the first event observed in the transition will be the synchronization of such nodes in a cluster, occurring at $d_1 = v^*/\lambda_8 = v^*/6$.

Then, one continues inspecting the matrices $S_n$ and concentrates only on all the other entries. No further entry is found equal to 2 in $S_7$, nor in $S_6$. When scrutinizing $S_5$ [Fig. 2d] one sees that other entries becomes equal to 2, and they indicate that the cluster formed by nodes $4, 5$ and $6$ will merge at $d_2 = v^*/\lambda_5 = v^*/4 = \frac{3}{2} d_1$ with the already existing group of synchronized nodes, forming this way a larger synchronized cluster. No further features are observed in $S_4$ and $S_3$, while the analysis of $S_2$ [Fig. 2e] reveals that at $d_3 = v^*/\lambda_2 = v^* = 4d_2$ also nodes $1, 2$ and $3$ join the existing cluster, determining the setting of the final synchronized state, where all nodes of the network evolve in unison.

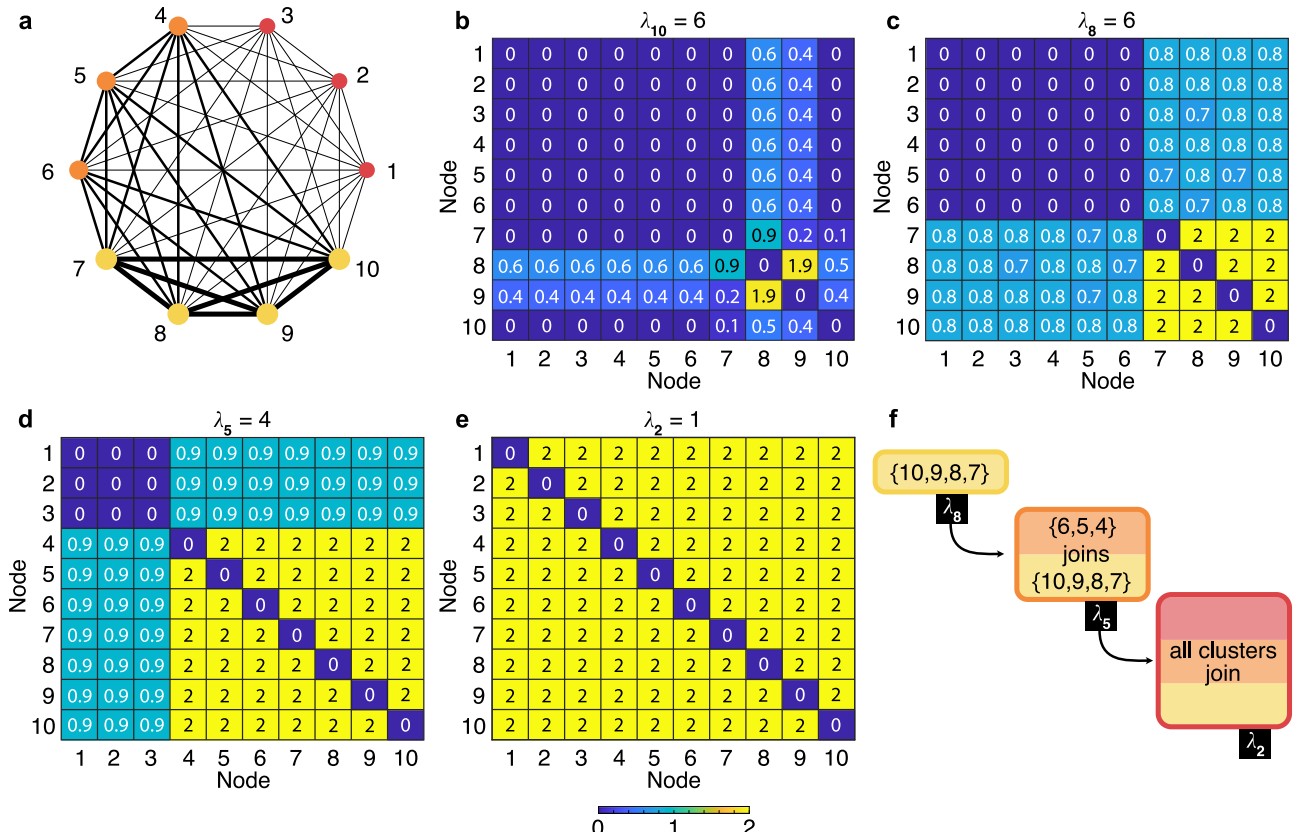

**Fig. 2 | Predicting the transition to synchronization. a** An all-to-all connected, symmetric, weighted graph of $N = 10$ nodes is considered. The graph is endowed with three symmetry orbits: the one composed by the red nodes $\{1, 2, 3\}$, the one made of the orange nodes $\{4, 5, 6\}$, and the one made of the four yellow nodes $\{7, 8, 9, 10\}$. In the sketch, the widths of the links are proportional to the corresponding weights, and the sizes of the nodes are proportional to the corresponding strengths. **b** The entries of $S_{10}$, which corresponds to $\lambda_{10} = 6$. **c** $S_8$ (associated to $\lambda_8 = 6$), where the entries equal to 2 clearly define a cluster formed by nodes $\{7, 8, 9, 10\}$. The first predicted event in the transition then consists in the synchronization of such nodes at $d_1 = v^*/\lambda_8 = v^*/6$. **d** $S_5$ (related to $\lambda_5 = 4$), where additional entries become equal to 2, indicating a second foreseen event in which nodes $\{4, 5, 6\}$ join the existent synchronization cluster at $d_2 = v^*/\lambda_5 = v^*/4$. **e** $S_2$ (corresponding to $\lambda_2 = 1$) where it is seen that nodes $\{1, 2, 3\}$ also join the existing synchronized cluster at $d_3 = v^*/\lambda_2 = v^*$ in a third predicted event where complete synchronization of the network takes place. **f** The expected events (and their exact sequence) occurring in the path to synchrony of the network's architecture depicted in **a**. The bar at the bottom of the Figure gives the color code used in panels (b-e) for matrices' entries.

Figure. 2f schematically summarizes the predicted sequence of events: if a dynamical system is networking with the architecture of Fig. 2a, its path to synchrony will first (at $d = d_1$) see the formation of the synchronized cluster $\{7, 8, 9, 10\}$, then (at $d = d_2$) nodes $\{4, 5, 6\}$ will join that cluster, and eventually (at $d = d_3$) all nodes will synchronize. Already at this stage it should be remarked that all predictions made are totally independent on **f** and **g**: changing the dynamical system **f** operating on each node and/or the output function **g** will result in the same sequence of events. The only difference will be that the estimated values $d_i$ at which the $i^{th}$ event will occur will be, under the adopted approximation, rescaled with the corresponding value of $v^*$.

In order to show how factual is our prediction, we monitored the transition in numerical simulations of Eq. (1), by using the Laplacian matrix of the network of Fig. 2a, and by considering two different dynamical systems: the Rössler[40] and the Lorenz system[41] with proper output functions. Precisely, the case of the Rössler system corresponds to $\mathbf{x} \equiv (x, y, z)$, $\mathbf{f}(\mathbf{x}) = (-y - z, x + ay, b + z(x - c))$ and $\mathbf{g}(\mathbf{x}) = (0, y, 0)$, while the case of the Lorenz system corresponds to $\mathbf{x} \equiv (x, y, z)$, $\mathbf{f}(\mathbf{x}) = (\sigma(y - z), x(\rho - z) - y, xy - \beta z)$ and $\mathbf{g}(\mathbf{x}) = (x, 0, 0)$. When parameters are set to $a = 0.1, b = 0.1, c = 18, \sigma = 10, \rho = 28, \beta = 2$, both systems are chaotic and belong to Class II, with $v^* = 7.322$ for the Lorenz case and $v^* = 0.179$ for the Rössler case. It is worth noticing that we here concentrate on chaotic systems, and do not consider instead periodic dynamics. The reason is that, in the MSF, $v = 0$ corresponds to $\lambda_1 = 0$ i.e., to the eigenvector $\mathbf{v}_1$ aligned with the synchronization manifold. Therefore, $\Lambda(0)$ is the maximum Lyapunov exponent of the isolated system which is equal to 0 for a periodic dynamics, leading to the vanishing of the critical value $v^*$ as well as of the threshold for complete synchronization. In turn, this would imply that intermediate clusters could not be observed as the entire network would synchronize for any infinitesimal coupling strength.

In order to properly quantify synchronization, one calculates the synchronization error over a given cluster, as

$$E_{cl} = \left\langle \left( \frac{1}{N_{cl}} \sum_i |\mathbf{x}_i - \bar{\mathbf{x}}_{cl}|^2 \right)^{\frac{1}{2}} \right\rangle_{\Delta T}, \qquad (3)$$

where the sum runs over all nodes $i$ forming the cluster, $\langle \ldots \rangle_{\Delta T}$ stands for a temporal average over a suitable time span $\Delta T$, $\bar{\mathbf{x}}_{cl}$ is the average value of the vector $\mathbf{x}$ in the cluster, and $N_{cl}$ is the number of nodes in the cluster. In addition, the synchronization error is normalized to its maximum value, so as to range from 1 to 0.

The results are reported in Fig. 3, where the normalized synchronization errors are reported as a function of $d$ (for both the Lorenz and the Rössler case) for the entire network (black dotted line) and for each of the three orbits in the network (yellow, orange and red lines). It is clearly seen that all predictions made are fully satisfied. Panels (b1-

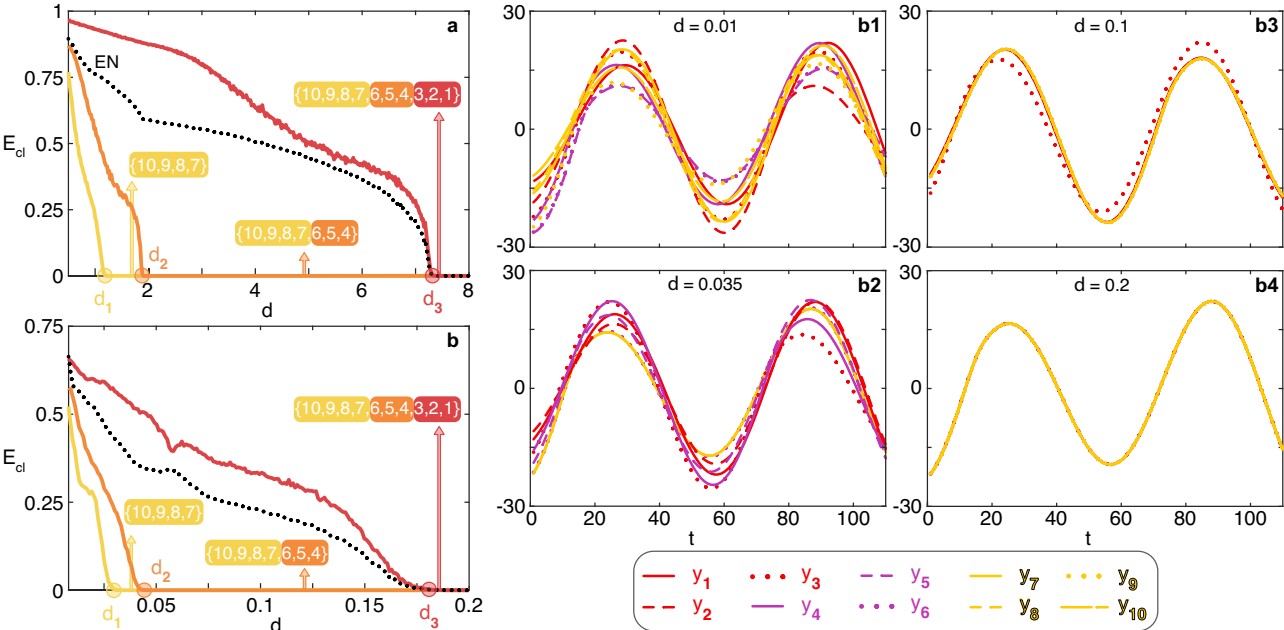

**Fig. 3 | The numerical verification of the predicted transition. a, b** The normalized synchronization errors $E_{cl}$ (see text for definition) as a function of $d$, for the Lorenz (**a**) and the Rössler (**b**) case. Data refers to ensemble averages over 500 different numerical simulations of the network sketched in (**a**) of Fig. 2. Cluster 1 (yellow line) is formed by nodes {7, 8, 9, 10}, Cluster 2 (orange line) is formed by nodes {4, 5, 6}, and Cluster 3 (red line) is formed by nodes {1, 2, 3}. The black dotted line refers to the synchronization error of the entire network (EN). In both panels it is seen that the expected sequence of events taking place during the transition is verified. Furthermore, the values $d_1 = 7.322/6 = 1.220$ ($d_1 = 0.179/6 = 0.0298$), $d_2 = 7.322/4 = 1.8305$ ($d_2 = 0.179/4 = 0.04475$) and $d_3 = 7.322$ ($d_3 = 0.179$) are marked in the horizontal axis respectively with a yellow, orange and red filled dot in (**a**) (in **b**)), indicating how accurate are the predictions and approximations made on the

corresponding critical values for the coupling strength. For each interval, the arrow points to the composition of the synchronized cluster that is being observed, once again in perfect harmony with the predictions made. Finally, we have verified that no extra synchronization features emerge during the transition, other than those explicitly foreseen in Fig. 2. b1–b4 Temporal snapshots illustrating the evolution of the $y$ variable of each of the 10 network's nodes (see color code at the bottom of the four panels) during the transition to synchronization reported in **b**. At $d = 0.01$ (b1) the nodes display a fully uncorrelated dynamics. At $d = 0.035$ b2 the yellow nodes (7,8,9,10) are clustered and display a synchronous motion, whereas all other nodes feature a uncorrelated dynamics. At $d = 0.1$ (b3) the violet nodes (4,5,6) have joined the clustered evolution, while nodes (1,2,3) remains unsynchronized. Finally, at $d = 0.2$ (b4), all network's nodes are synchronized.

b4) of Fig. 3 report temporal snapshots of the dynamics of each of the 10 network's nodes, and illustrate visually the different collective phases which are observed during the transition to synchronization reported in Fig. 3b.

Looking at Fig. 3, it must be remarked that the transition to synchronization is identical for the Rössler and Lorenz systems, with the only difference being the different values $v^*$ (0.179 for Rössler and 7.322 for Lorenz). In other words, the horizontal axis in Fig. 3b just corresponds to that of Fig. 3a multiplied by the ratio 0.179/7.322 of the two critical values. This depends on the fact that the set of clusters that are synchronizing during the transition, the nodes that compose them, and the order in which clusters synchronize are, under the approximation adopted in our study, completely independent on the specific dynamical system that is ruling the evolution of the network's nodes (as these features only depend instead on the spectrum of the Laplacian matrix, and consequently only on the topology of the graph). This is a strong result, because it implies that in all practical situations (i.e., when there are uncertainties in the model parameters, or even when the knowledge of the dynamics of the nodes is fully unavailable) the entire cluster sequence forming the transition to synchronization can still be predicted.

## Synthetic networks of large size

The next step is to test the accuracy of the method in the case of large size graphs. To this purpose, we consider 2 networks that were synthetically generated in ref. 42, where a specific method of generating graphs endowed with desired clusters was introduced that initially considers ensembles of disconnected nodes and then connects each

one of the nodes in each of such ensembles to the same group of nodes of an external core network, this way forming clusters containing nodes of equal degree. While we refer the reader to ref. 42 for the full details of the generating algorithm, we here limit ourselves to report the main attributes of the considered networks. The first network $G_1$ is of size $N = 1000$ nodes, and is endowed with two symmetry orbits that generate two distinct clusters of sizes 20 nodes (Cluster 1) and 10 nodes (Cluster 2), respectively. The second network $G_2$ is made of $N = 10,000$ nodes and is endowed with four symmetry orbits generating four clusters of sizes that span more than an order of magnitude (Clusters 1 to 4 contain, respectively, 1000, 300, 100, and 30 nodes).

Following the expectations which are detailed in the theorems of our Supplementary Information, the calculation of the Laplacian eigenvalues of $G_1$ allows one to identify a first group of 19 degenerate eigenvalues $\lambda_{277,278,...,295} = 4$ and a second group of 9 degenerate eigenvalues $\lambda_{59,60,...,67} = 2$. The analysis of the matrices $S_n$ then reveals that the first event in the transition is the synchronization of Cluster 1 at $d_1 = v^*/4$, followed by the synchronization of Cluster 2 at $d_2 = v^*/2 = 2d_1$, this time in a state which is not synchronized with cluster 1, thus determining an overall state where two distinct synchronization clusters coexist. Eventually, the entire network synchronizes at $d_3 = v^*/\lambda_2 = v^*/0.4758$.

In the case of $G_2$, four blocks of degenerate eigenvalues are indeed found in correspondence with the four clusters. The transition predicted by inspection of the matrices $S_n$ is characterized by the sequence of five events. First, the cluster with 30 nodes synchronizes at $d_1 = v^*/5$. Second, at $d_2 = v^*/4$, the cluster with 100 nodes

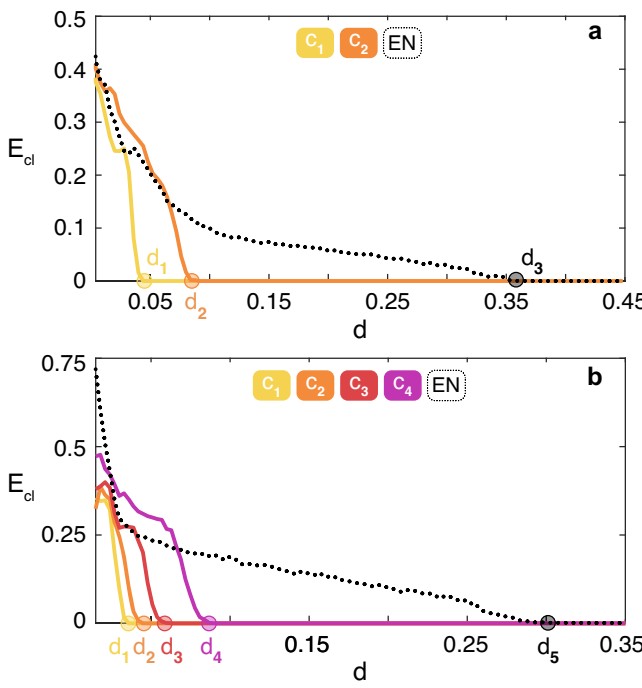

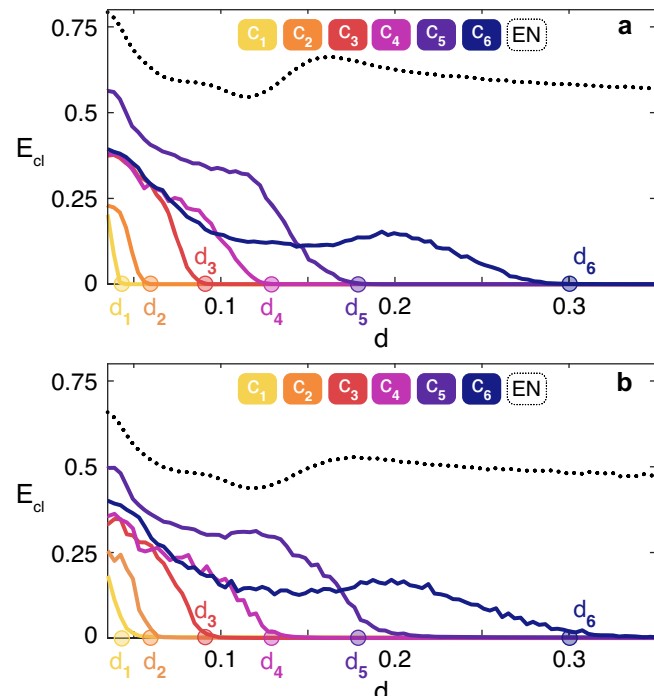

**Fig. 4 | Applications to large size synthetic networks.** $E_{cl}$ (see text for definition) vs. $d$, for the Rössler system (see the differential equations in the text). Data in **a** [in **b**] refer to ensemble averages over 50 (150) different numerical simulations of the graph $G_1$ ($G_2$) described in the main text. In both panels, the legend sets the color code for the curves corresponding to each of the existing clusters $C_i$ and to the Entire Network (EN). Once again, the two predicted transitions are verified. **a** Cluster 1 synchronizes at $d_1 = 0.179/4 = 0.04475$ (marked with a yellow dot), Cluster 2 synchronizes at $d_2 = 0.179/2 = 0.0895$ (orange dot), and the entire network synchronizes at $d_3 = 0.179/0.4758 = 0.376$ (black dot), as predicted. **b** The cluster with 30 nodes synchronizes at $d_1 = 0.179/5 = 0.0358$ (yellow dot), the cluster with 100 nodes at $d_2 = 0.179/4 = 0.04475$ (orange dot), the cluster with 300 nodes at $d_3 = 0.179/3 = 0.0597$ (red dot), the cluster with 1,000 nodes at $d_4 = 0.179/2 = 0.0895$ (violet dot), and the entire network at $d_5 = 0.179/0.6025 = 0.297$ (black dot).

**Fig. 5 | Applications to the PowerGrid network.** $E_{cl}$ (see text for definition) vs. $d$, for the Rössler system (see the differential equations in the text). Data in **a** [in **b**] refer to ensemble averages over 850 (200) different numerical simulations of the PowerGrid network. As in Fig. 4, the legends of both panels set the color code for the curves corresponding to each of the reported clusters $C_i$ and to the Entire Network (EN). **a** reports the case of identical Rössler systems, and the error of 6 specific clusters is plotted (see the Supplementary Information for the composition of each of the 6 clusters $C_i$). The observed sequence of events perfectly matches the predicted one, with an excellent fit with the values $d_1, \ldots, d_6$. In panel (**b**) the effects of heterogeneity in the network are reported. Namely, for each node $i$ of the PowerGrid network, the parameter $b_i$ in the Rössler equations is randomly sorted from a uniform distribution in the interval $[0.1 - \epsilon, 0.1 + \epsilon]$. The curves plotted refer to $\epsilon = 0.01$.

synchronizes. At $d_3 = v^*/3$ ($d_4 = v^*/2$) also the cluster with 300 nodes (with 1000 nodes) synchronizes. The four clusters evolve in four different synchronized states. Eventually, at $d_5 = v^*/0.6025$ the entire network synchronizes.

We then simulated the Rössler system on $G_1$ and $G_2$ and reported the results in Fig. 4, which are actually fully confirming the predicted scenarios.

**Real-world and heterogeneous networks**

We move now to show three applications to real-world networks.

For the first application, we have considered the network of the US power grid. The PowerGrid network consists of 4,941 nodes and 6,594 links, and it is publicly available at https://toreopsahl.com/datasets/#uspowergrid. It was already the object of several studies in the literature, the first of which was the celebrated 1998 paper on small world networks[43]. In the PowerGrid network, nodes are either generators, transformers or substations forming the power grid of the Western States of the United States of America, and therefore they have a specific geographical location. Recently, it was proven that a fraction of 16.7% of its nodes are forming non trivial clusters corresponding to symmetry orbits which are small in size, due to the geographical embedding of the graph[26].

Application of our method detects that the synchronization transition is made of a very well defined sequence of events, which

involves the emergence of 381 clusters. The clusters that are being formed are all small in size, because of the constraints made by the geographical embedding. In particular, 310 clusters contain only 2 nodes, 49 clusters are made of 3 nodes, 14 clusters are formed by 4 nodes, 4 clusters have 5 nodes, 2 clusters appear with 6 nodes, 1 cluster has 7 nodes, and 1 cluster is made of 9 nodes. The overall number of network's nodes which get clustered during the transition is 871. A partial list of these clusters (spanning about one order of magnitude in the size of the corresponding eigenvalues) and the various values of coupling strengths at which the different events are predicted is available in Table 1 of Supplementary Information.

We have then simulated the Rössler system on the PowerGrid network, and monitored the synchronization error on 6 specific clusters (highlighted in red in the list of Table 1 of Supplementary Information) that our method foresees to emerge during the path to synchrony in a well established sequence and at well specific values of the coupling strength ($d_1 = 0.04475$, $d_2 = 0.0596$, $d_3 = 0.0895$, $d_4 = 0.1294$, $d_5 = 0.179$, $d_6 = 0.3056$). The values $d_1, \ldots, d_6$ are explicitly calculated in the Supplementary Information, and are marked as filled dots in the horizontal axis of Fig. 5 with the same colors identifying the corresponding clusters. Looking at Fig. 5a one sees that, once again, the observed sequence of events matches the predicted one, with an excellent fit with the values $d_1, \ldots, d_6$, thus validating ex-post the approximation adopted in our study.

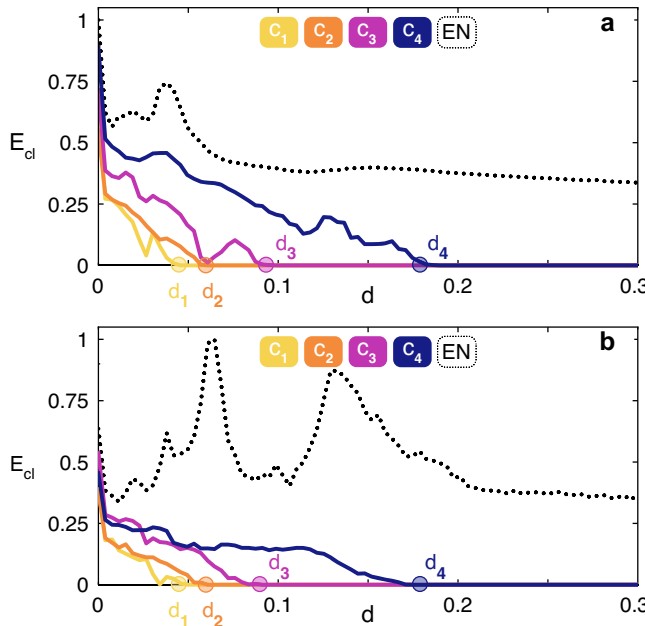

**Fig. 6 | Application to biological and social networks.** Synchronization error for the considered clusters $C_1$, $C_2$, $C_3$ and $C_4$ (see the color-code at the top of the panel) for the Yeast protein-protein interaction network[44] (**a**), and the ego-Facebook network[45] **b**. In both panels, we report also the synchronization error of the entire network (EN, black dotted line).

We also tested how robust is the predicted scenario against possible heterogeneities present in the network. To this purpose, one again simulates the Rössler system on the PowerGrid network, but this time one distributes randomly different values of the parameter $b$ to different network's nodes. Precisely, for each node $i$ of the network, one uses a parameter $b_i$ in the Rössler equations which is randomly sorted from a uniform distribution in the interval $[0.1 - \epsilon, 0.1 + \epsilon]$, with the extra parameter $\epsilon$ that now quantifies the extent of heterogeneity in the graph. The results are reported in panel (b) of Fig. 5 for $\epsilon = 0.01$, corresponding to 10% of the value ($b = 0.1$) which was used for all nodes in the case of identical systems, thus representing a case of a rather large heterogeneity.

It has to be remarked that, when the networked units are not identical, the very same synchronization solution ceases to exist, and therefore it formally makes no sense to speak of the stability of a solution that does not even exist. Indeed, if one selects no matter which ensemble of nodes, the synchronization error never vanishes exactly in the ensemble. Nonetheless, it is still observed that the values of the normalized synchronization errors fluctuate around zero for some sets (clusters) of network's nodes which, therefore, anticipate the setting of the almost completely synchronized state (wherein all nodes evolve almost in unison). In Fig. 5b it is clearly seen that, while the synchronization errors approach zero at values that are obviously different from those predicted in the case of identical systems, the sequence at which the different clusters emerge during the transition is still preserved. Similar scenarios characterize the evolution of the network also when the heterogeneity is affecting the other two parameters ($a$ and $c$) entering into the equations of the Rössler system.

The second (third) application is given with reference to a real-world biological (social) network. Precisely, we consider the Yeast protein-protein interaction network (a dataset made of $N = 1647$ nodes and $E = 2,518$ edges[44]) and the ego-Facebook network (a dataset containing $N = 2888$ nodes and $E = 2981$ edges[45]). The results are reported in Fig. 6, and refer to ensemble averages over 100 different numerical simulations of identical Rössler systems (same

parameters and initial conditions as in the case reported in Fig. 5) coupled by the structure of connections of the two graphs. For the biological network, it is found that the transition to synchronization includes 188 clusters, and Fig. 6a reports the synchronization error of 4 of them: $C_1$ and $C_2$ (which are both 2 nodes clusters), $C_3$ (a cluster containing 3 nodes), $C_4$ (a cluster of 7 nodes). For the social network, 18 clusters are found during the transition to synchronization, and also in this case Fig. 6b reports the synchronization error of 4 of them: $C_1$ and $C_2$ (which are both made of 2 nodes), $C_3$ (a cluster containing 5 nodes), $C_4$ (a subset of 3 nodes−out of a cluster of 280 nodes−which is forming a cluster by itself). In both cases, one easily sees that the numerically observed sequence of events perfectly matches the predicted one, with an excellent fit of the critical coupling strength values $d_1$, $d_2$, $d_3$, and $d_4$.

## Discussion

In conclusion, our work provides an approximated, yet very accurate, description of the transition to synchronization in a network of identical systems. We unveil, indeed, that the path to synchronization is made of a sequence of events, each of which can be identified as either the nucleation of one (or several) cluster(s) of synchronized nodes, or to the merging of multiple synchronized clusters into a single one, or to the growth of an already existing synchronized cluster which enlarges its size. In our study, all nodes in a cluster have the same connections (and the same weights) with nodes not belonging to the cluster, and therefore they receive the same dynamical input from the rest of the network.

By combing methodologies borrowed from stability of nonlinear systems with tools of algebra and symmetries, and under the approximation that the systems' trajectories in all cluster-synchronous states do not substantially differ from those featured at complete synchronization, we have been able to introduce a simple and effective method able to provide the complete prediction of the sequence of such events, to identify which graph's node is belonging to each of the emergent clusters, and to give a forecast of the critical coupling strength values at which such events are taking place.

While the local node dynamics and the coupling function are entirely responsible for the specific synchronization class of the Master Stability Function, the sequence of events that are taking place during the transition to synchronization depends, instead, only on the graph's structure and, more precisely, on the only knowledge of the full spectrum of eigenvalues and eigenvectors of the Laplacian matrix.

Our method does not require an a-priori knowledge of the network's symmetries. We moreover clarify once forever the intimate nature of the clusters that are being formed along the transition path: they are formed by those nodes which are indistinguishable at the eyes of any other network's vertex. This implies that nodes in a synchronized cluster have the same connections (and the same weights) with nodes not belonging to the cluster, and therefore they receive the same dynamical input from the rest of the network. Synchronizable clusters in a network are therefore subsets more general than those defined by the graph's symmetry orbits, and at the same time more specific than those described by equitable partitions.

Finally, our work gave evidence of several extensive numerical simulations with both synthetic and real-world networks, and demonstrates how high is the accuracy of our predictions. Remarkably, the synchronization scenario in heterogeneous networks (i.e., networks made of non-identical units) preserves the predicted cluster sequence along the entire synchronization path.

Our results, therefore, call for a lot of applications of general interest in nonlinear science, ranging from synthesizing networks equipped with desired cluster(s) and modular behavior, until predicting the parallel (clustered) functioning of real-world networks from the analysis of their structure.

## Methods

In what follows we consider a connected weighted undirected graph $G$ with $N$ nodes and uniquely identified by its adjacency matrix $A$ and its Laplacian matrix $\mathcal{L}$. Furthermore, we will call $\lambda_1 = 0 < \lambda_2 \leq \lambda_3 \ldots \leq \lambda_N$ the (ordered in size) real and nonnegative eigenvalues of $\mathcal{L}$, and $\mathbf{v}_1 = \frac{1}{\sqrt{N}}(1,1,1,\ldots,1)^T$, $\mathbf{v}_2, \mathbf{v}_3, \ldots, \mathbf{v}_N$ corresponding orthonormal eigenvectors.

### The MSF for the network's synchronous solution

Let us recall here that the equation governing a generic ensemble of $N$ identical dynamical systems interplaying over a network $G$ is Eq. (1) i.e., $\dot{\mathbf{x}}_i = \mathbf{f}(\mathbf{x}_i) - d \sum_{j=1}^{N} \mathcal{L}_{ij} \mathbf{g}(\mathbf{x}_j)$, where $\mathbf{x}_i(t)$ is a $m$-dimensional vector state describing the dynamics of each node $i$, $\mathbf{f} : \mathbb{R}^m \longrightarrow \mathbb{R}^m$ is the local (identical in all units) dynamical flow describing the evolution of the isolated systems, $d$ is a real-valued coupling strength, $\mathcal{L}_{ij}$ is the $ij$ entry of the Laplacian matrix, and $\mathbf{g} : \mathbb{R}^m \longrightarrow \mathbb{R}^m$ is an output function describing the functional way through which units interplay.

$\mathcal{L}$ is a zero-row matrix, a property which, in turn, guarantees existence and invariance of the synchronized solution

$$\mathbf{x}_s(t) = \mathbf{x}_1(t) = \ldots = \mathbf{x}_N(t).$$

In order to study stability of such a solution, one considers perturbations $\delta \mathbf{x}_i = \mathbf{x}_i - \mathbf{x}_s$ for $i = 1, \ldots, N$, and performs linear stability analysis of Eq. (1). The result are the following equations:

$$\dot{\delta \mathbf{x}}_i = J\mathbf{f}(\mathbf{x}_s)\delta \mathbf{x}_i - d\sum_{j=1}^{N} \mathcal{L}_{ij} J\mathbf{g}(\mathbf{x}_s)\delta \mathbf{x}_j, \tag{4}$$

where $J\mathbf{f}(\mathbf{x}_s)$ and $J\mathbf{g}(\mathbf{x}_s)$ are, respectively, the $m \times m$ Jacobian matrices of the flow and of the output function, both evolving in time following the synchronization solution's trajectory.

One can, in fact, consider the global error $\delta \mathbf{X} \in \mathbb{R}^{Nm} \equiv (\delta \mathbf{x}_1, \delta \mathbf{x}_2, \ldots, \delta \mathbf{x}_N)^T$ around the synchronous state, and recast Eq. (4) as

$$\dot{\delta \mathbf{X}} = \left[ \mathbb{I} \otimes J\mathbf{f}(\mathbf{x}_s) - d\mathcal{L} \otimes J\mathbf{g}(\mathbf{x}_s) \right] \delta \mathbf{X}, \tag{5}$$

where $\mathbb{I}$ is the identity matrix, and $\otimes$ stands for the direct product.

Moreover, $\mathcal{L}$ is a symmetric, zero row sum, matrix. As so, it is always diagonalizable, and the set of its eigenvectors forms an orthonormal basis of $\mathbb{R}^N$. The zero row sum property of $\mathcal{L}$ implies furthermore that $\lambda_1 = 0$ and that $\mathbf{v}_1 = \frac{1}{\sqrt{N}}(1,1,\ldots,1,1)^T$. Therefore, all components of $\mathbf{v}_1$ are equal, and this means that $\mathbf{v}_1$ is aligned, in phase space, to the manifold $\mathcal{M}$ defined by the synchronization solution, and that an orthonormal basis for the space $\mathcal{T}$ tangent to $\mathcal{M}$ is just provided by the set of eigenvectors $\mathbf{v}_2, \mathbf{v}_3, \ldots, \mathbf{v}_N$. For the synchronization solution to be stable, the necessary condition is then that all directions of the tangent space be contractive.

One can now expand the error $\delta \mathbf{X}$ as a linear combination of the eigenvectors $\{\mathbf{v}_i\}$ i.e.,

$$\delta \mathbf{X} = \sum_{i=1}^{N} \mathbf{v}_i \otimes \boldsymbol{\eta}_i.$$

Then, plugging the expansion in Eq. (5) and operating the scalar product of both the right and left part of the equation times the eigenvectors $\mathbf{v}_i$, one obtains that the coefficients $\boldsymbol{\eta}_i \in \mathbb{R}^m$ obey the equations

$$\dot{\boldsymbol{\eta}}_i = \left[ J\mathbf{f}(\mathbf{x}_s) - d\lambda_i J\mathbf{g}(\mathbf{x}_s) \right] \boldsymbol{\eta}_i.$$

Notice that the equations for the coefficient $\boldsymbol{\eta}_i$ are variational, and only differ (at different $i's$) for the eigenvalue $\lambda_i$ appearing in the evolution kernel. This entitles one to cleverly separate the structural and dynamical contributions, by introducing a parameter $\nu \equiv d\lambda$, and

by studying the $m$-dimensional parametric variational equation

$$\dot{\boldsymbol{\eta}} = \left[ J\mathbf{f}(\mathbf{x}_s) - \nu J\mathbf{g}(\mathbf{x}_s) \right] \boldsymbol{\eta} = K(\nu)\boldsymbol{\eta}. \tag{6}$$

The kernel $K(\nu)$, indeed, only depends on $\mathbf{f}$ and $\mathbf{g}$ (i.e., on the dynamics), and the structure of the network is now encoded within a specific set of $\nu$ values (those obtained by multiplying $d$ times the Laplacian's eigenvalues).

The maximum Lyapunov exponent $\Lambda$ [i.e., the maximum of the $m$ Lyapunov exponents of Eq. (6)] can then be computed for each value of $\nu$. The function $\Lambda(\nu)$ is called the Master Stability Function, and only depends on $\mathbf{f}$ and $\mathbf{g}$. At each value of $d$, a given network architecture is just mapped to a set of $\nu \neq 0$ values. The corresponding values of $\Lambda(\nu)$ provide the expansion (if positive) or contraction (if negative) rates in the directions of the eigenvectors $\mathbf{v}_2, \mathbf{v}_3, \ldots, \mathbf{v}_N$, and therefore one needs all these values to be negative in order for $\mathcal{M}$ to be attractive in all directions of $\mathcal{T}$.

Finally, notice that $\nu = 0$ corresponds to $\lambda_1 = 0$ i.e., to the eigenvector $\mathbf{v}_1$ aligned with $\mathcal{M}$. Therefore, $\Lambda(0)$ is equal to the maximum Lyapunov exponent of the isolated system $\dot{\mathbf{x}} = \mathbf{f}(\mathbf{x})$. In turn, this implies that the Master Stability Function starts with a value which is strictly positive (strictly equal to 0) if the networks units are chaotic (periodic).

The 3 different Classes of systems supported by the Master Stability Function are illustrated in Fig. 1, and largely discussed in the Manuscript.

### The stability properties of the clusters' synchronous solution

It is important to remark that all the above results are formally valid only for the whole network's synchronous solution. The trajectories followed by the nodes' dynamics in each cluster-synchronous state slightly differ, instead, from those which are followed in the global solution, as they rigorously depends on the quotient network, and therefore on topology, node dynamics, and clusterization. Let us indeed focus on a given cluster $C_l$, and let us call $\mathcal{C}_l$ the set of indices identifying the nodes that belong to $C_l$. For each node $i$ ($i \in \mathcal{C}_l$), Eq. (1) becomes

$$\dot{\mathbf{x}}_i = \mathbf{f}(\mathbf{x}_i) - d\sum_{j \in \mathcal{C}_l} \mathcal{L}_{ij} \mathbf{g}(\mathbf{x}_j) - d\sum_{j \notin \mathcal{C}_l} \mathcal{L}_{ij} \mathbf{g}(\mathbf{x}_j), \tag{7}$$

where the overall coupling is now split into the sum of an intra-cluster term and of a term accounting for the connections of the cluster to the rest of the network. Eq. (7) can be rewritten as

$$\dot{\mathbf{x}}_i = \mathbf{f}(\mathbf{x}_i) - d\sum_{j \in \mathcal{C}_l} \mathcal{L}_{ij} \mathbf{g}(\mathbf{x}_j) + d\sum_{j \notin \mathcal{C}_l} \mathcal{A}_{ij} \mathbf{g}(\mathbf{x}_j), \tag{8}$$

where $\mathcal{A}$ is the adjacency matrix. This is because all the elements of the Laplacian matrix considered in the second coupling term are just the opposite of the corresponding terms of the adjacency matrix. The second coupling term is, indeed, limited to $j \notin \mathcal{C}_l$ and therefore, by definition, it does not contain the diagonal element of the Laplacian ($j = i$) which is instead contained in the intra-cluster coupling term.

We now recall that the main theorem of our study asserts that synchronizable clusters are those formed by nodes which are equally connected to, and receive an equal input from, the rest of the network. Therefore, as the last term of Eq. (8) accounts for the total input received by node $i$ from all nodes that do not belong to the cluster, our theorem ensures that it is a term which is formally independent on $i$. The cluster synchronous solution is $\mathbf{x}_i(t) = \mathbf{x}_j(t) = \mathbf{x}_{C_l}(t), \forall i \in \mathcal{C}_l$ and $\forall j \in \mathcal{C}_l$ ($j \neq i$), and obeys the equation

$$\dot{\mathbf{x}}_{C_l} = \mathbf{f}(\mathbf{x}_{C_l}) + d\sum_{j \notin \mathcal{C}_l} \mathcal{A}_{ij} \left[ \mathbf{g}(\mathbf{x}_j) - \mathbf{g}(\mathbf{x}_{C_l}) \right]. \tag{9}$$

This is because the diagonal terms of the Laplacian matrix are equal to the sum of the number of intra-cluster connections and of the number of extra-cluster connections, the latter ones being now incorporated in the remaining coupling term, which once again does not depend on $i$. Once again, one can consider perturbations $\delta\mathbf{x}_i = \mathbf{x}_i - \mathbf{x}_{C_l}$ ($\forall i \in C_l$), and perform linear stability analysis of Eq. (8). The result is

$$\dot{\delta\mathbf{X}} = \left[ \mathbb{I} \otimes J\mathbf{f}(\mathbf{x}_{C_l}) - d\mathcal{L} \otimes J\mathbf{g}(\mathbf{x}_{C_l}) \right] \delta\mathbf{X}, \qquad (10)$$

where $\delta\mathbf{X} \in \mathbb{R}^{N_l m}$ is the global error vector, and $N_l$ is the number of nodes forming the cluster $C_l$.

It is immediately seen that the linearized Eq. (10) is formally identical to Eq. (5), and therefore the same expansion of the error can be made with the eigenvectors of the Laplacian. The difference, however, is that the evaluation of the maximum Lyapunov exponents requires now to calculate the Jacobians of the functions $\mathbf{f}$ and $\mathbf{g}$ over the cluster-synchronous solution $\mathbf{x}_{C_l}$, which obeys Eq. (9). In other words, while the MSF formalism calculates the Maximum Lyapunov exponents using the trajectories experienced by the whole network's synchronous solution (a state in which each node of the network repeats the same dynamical behavior of a single, isolated, system), the trajectories experienced by the cluster synchronous state are perturbed by an extra term $K(t) = d\sum_{j \notin C_l} \mathcal{A}_{ij} [\mathbf{g}(\mathbf{x}_j) - \mathbf{g}(\mathbf{x}_{C_l})]$, and depend therefore explicitly on the entire network's dynamics, and on the specific coupling function. This fact leads to two main consequences:

- The very same cluster-synchronous solution is not invariant, as $K(t)$ explicitly depends on $d$. In particular, at each value of the coupling strength one would have a distinct cluster-synchronous state, and therefore the entire framework of the MSF would make no sense in this case as it would not be possible to rigorously separate dynamics and structure;
- The perturbation $K(t)$ may lead the synchronous trajectories to visit areas of the phase space which are instead never visited by a single isolated system, and therefore it may determine slight variations in the calculations of the maximum Lyapunov exponents, and consequently slight variations in the determination of the critical coupling strength value at which the cluster synchronous state becomes stable.

On the other hand, the term $K(t)$ is directly proportional to $d$, and therefore it has to be expected that such a perturbation will in fact be small across the transition to complete synchronization, where $d$ starts from 0 and only slightly increases to values which are normally smaller than 1. In addition, $K(t)$ consists of the sum of terms which are in general uncorrelated, as there are no constraints on the dynamics of the nodes which do not form part of the synchronous cluster. This latter fact would also contribute to determine smallness of the perturbation.

In our study, we decided therefore to adopt a practical approximation to the problem, by assuming that the perturbation $K(t)$ is always negligible and consequently that the trajectories visited by the clustered synchronous nodes are always equal to those that characterize complete synchronization (i.e., those of a single, isolated, system). This allows one to refer to a unique MSF (the one constructed in the whole network's synchronous state) for determining the stability properties of all cluster synchronized states.

## Simulations

Simulations were performed with an adaptive Tsit integration algorithm implemented in Julia. In each trial, the network is simulated for a total period of 1,500 time units, and the synchronization errors are averaged over the last $\Delta T = 100$ time units.

As one is only interested to monitor the vanishing of $E_{cl}$, with the purpose of saving calculations in all our simulations the synchronization error is computed by only taking into account those variables of the system's state where the coupling is acting. This implies that, when referring to the Rössler (Lorenz) system, $E_{cl}$ has been evaluated taking into account only the difference $y_i - \bar{y}_{cl}$ ($x_i - \bar{x}_{cl}$). The results are, indeed, identical to those obtained when all state variables of the systems are accounted for in the evaluation of $E_{cl}$, as its formal definition of Eq. (3) would instead require.

## Reporting summary

Further information on research design is available in the Nature Portfolio Reporting Summary linked to this article.

## Data availability

The data supporting the findings of this study are available within the article and Supplementary Information, and can be accessed in the repository https://github.com/goznalo-git/ClusterSynchronization.

## Code availability

All relevant codes used are open source and available at the repository https://github.com/goznalo-git/ClusterSynchronization.

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

## Acknowledgements

G.C.-A. and K.A.-B. acknowledge funding from projects M1993 and M2978 (URJC Grants). G.C.-A. acknowledges funding from the URJC fellowship PREDOC-21-026-2164. S.B. acknowledges support from the project n.PGR01177 of the Italian Ministry of Foreign Affairs and International Cooperation.

## Author contributions

S.B. and S.J. designed the research project. A.B., F.N., G.C.-A. and K.A.-B. performed the simulations, K.K. and R.S.-G. developed the mathematical formalism. S.B. and S.J. jointly supervised the research. All authors jointly wrote and reviewed the manuscript.

## Competing interests

The authors declare no competing interests.
