## [Peer Review File · Nature Communications]

REVIEWER COMMENTS

Reviewer #1 (Remarks to the Author):

This paper is concerned with a numerical method to accurately predict the sequence of events (emergence of synchronized clusters) that take place in a networked system during its transition to global (non-explosive) synchronization.

The first main drawback of this paper is that the Master Stability Function approach is based on the assumption of infinitesimal perturbation about a synchronous solution. Therefore, the stability of any cluster synchronization pattern depends on the related quotient network and on its dynamics, as shown in many works by Pecora et al. (e.g., Sorrentino, F., Pecora, L. M., Hagerstrom, A. M., Murphy, T. E., & Roy, R. (2016). Complete characterization of the stability of cluster synchronization in complex dynamical networks. *Science Advances*, 2(4), e1501737 and Siddique, A. B., Pecora, L., Hart, J. D., & Sorrentino, F. (2018). Symmetry-and input-cluster synchronization in networks. *Physical Review E*, 97(4), 042217). This seems to be completely neglected by the authors.

Further weaknesses:

- The network model is not so general: homogeneous, undirected, and Laplacian network, absence of delays, output function depending only on x_j . The class of considered networks should be better evidenced from the beginning of the paper, maybe also in its title.
- The proposed results are concerned with networks with unrealistic dynamics, so there is no benchmark for the proposed results. Referring to (and making comparisons with) known results about real networks would make the paper much stronger. Talking about a “real-world network” for a network with real topology but unreal dynamics is misleading, in my opinion.
- A similar approach was already proposed in the following paper, not even cited by the authors: Fu, C., Lin, W., Huang, L., & Wang, X. (2014). Synchronization transition in networked chaotic oscillators: The viewpoint from partial synchronization. *Physical Review E*, 89(5), 052908.
- The plots shown in Fig. 1 are correct if one considers global synchronization in Laplacian networks. But for cluster synchronization, each graph should change according to the considered synchronous clusters and to the network topology. How is this accounted for in the proposed method? This is related to the general drawback pointed out above.
- In all the proposed examples the node dynamics are chaotic, even if the literature numbers many examples of analysis of networks whose node dynamics are periodic. At least one example of this kind would be useful, mainly in the presence of intertwined clusters. In any case, additional evidence is needed to support the main claims of the paper.

- The section titled “The synchronized solution” contains only known results and could be moved to the Supplementary Information.
- Condition 1 in the proposed Theorem is claimed to be “more specific than the graph’s equitable partitions”, but in several other papers the same condition is already used to find equitable clusters, e.g.
 - o [13] M. T. Schaub, N. O’Clery, Y. N. Billeh, J.-C. Delvenne, R. Lambiotte, and M. Barahona, “Graph partitions and cluster synchronization in networks of oscillators,” *Chaos: An Interdisciplinary Journal of Nonlinear Science*, vol. 26, no. 9, p. 094821, 2016.
 - o [14] A. B. Siddique, L. Pecora, J. D. Hart, and F. Sorrentino, “Symmetry-and input-cluster synchronization in networks,” *Physical Review E*, vol. 97, no. 4, p. 042217, 2018.
- There are some typos, e.g., “cluterize”, “The only different will be”.

Reviewer #2 (Remarks to the Author):

In this paper the authors study a rather general class of coupled dynamical systems operating on the nodes of complex networks. The dynamical process involves both an on-site dynamics (that can be possibly be chaotic) as well as a pairwise coupling between nodes, mediated by the graph Laplacian matrix.

The focus is on analyzing the progressive synchronization process, which emerges as an overall coupling-strength parameter, "d", is increased. For this, the authors rely on the well-known formalism of the Master Stability function(MSF). This allows them to separate the diverse "modes" associated with each of the eigenvalues of the Laplacian matrix and study their stability independently from each other. As d is increased a series of local synchronization events progressively unfolds following the sequence of Laplacian modes that become sequentially stabilized (i.e. their associated eigenvalues cross the stability limit). Eventually, the completely synchronized state becomes stabilized.

I would say that such a setup ---allowing to describe in terms of features of the Laplacian graph the cascade or partial

synchronization events leading eventually to global synchronization--- is very well known and well established in the literature.

The main novelty of the present manuscript is that it considers the specific case of networks with certain symmetries, i.e., assuming the existence of groups of nodes that can be permuted between themselves

without affecting their effect on the rest of the network, or in other words, they are indistinguishable nodes (to the "eyes of any other node of the network", to use the author's wording). Such groups of indistinguishable nodes are called "clusters". The main contribution of the manuscript is proving mathematical theorem, by virtue of which if a cluster has k nodes, then there is a spectral block defined by $k-1$ Laplacian eigenmodes, perfectly localized at exactly those k nodes.

A direct consequence of the theorem is that, for networks with the aforementioned clusters, the sequence of eigenvalues crossing the stability limit as " d " is increased, can be associated with an easily interpretable description of the

cascade of synchronization events. As a matter of fact, as the corresponding eigenvectors can be localized in clusters, the synchronization unfolds by progressively generating local synchronous clusters and then merging them.

The authors design a simple and clear protocol to actually do this and identify the synchronizing clusters in a progressive way and apply it to a simple toy network with some symmetries. The algorithm is verified to work fine both for small and for large networks provided that they have clusters (otherwise, there is no prediction to be made).

Finally, the method is also applied to a real example: the US power grid network. The authors detect a well-defined series of synchronization events and clusters and analyze how robust is the synchronization cascade when some degree of heterogeneity is added.

My main concern is about the actual relevance of the findings for real-world applications/networks. As I already wrote, the description of the unfolding of synchronization exploiting the properties of the Laplacian matrix is already well known, so the really new contribution of this work is that of extending those old ideas to networks with clusters. Thus, my feeling is that the authors should make a much stronger case justifying that clusters are generically present in real-world networks and show some other example (e.g. brain networks?) complementing the power grid one, to really convince the reader of the power of identifying clusters and applying the here-developed tools to uncover the route to synchronization in real-world networks.

In summary, I think that the paper is clear, sound and well-written. It certainly introduces new ideas and should be published in some form or the other. But depending on how strongly can the authors defend the importance of clusters for real-world networks. I would either recommend accepting the paper for

Nat. Comm. or either suggest it to be considered (accepted) in a less generalistic venue such as Communications Physics.

Minor comments:

- I believe there is a mistake/misprint in the first network of Fig.2. as entry (7,10) does not coincide with (10,7).

- I could not find the values of the weights in the discussed example. I think it is important to include them so that the reader can reproduce the calculation in detail. Moreover, it is important to further underline that these weights do not need to take the same

values within a cluster. E.g. in the discussed example it seems to me that nodes 8 and 9 are more stringly connected between themselves than e.g. 7 and 10, even if they all are within the same cluster.

- I would like the authors to discuss what happens if the dynamical process does not explicitly involve a Laplacian matrix for the coupling between nodes but, e.g., an adjacency one. In other words, the Laplacian already implies some conservation law that might be too restrictive for many real-world potential applications.

- Honestly, I think that the number of self-citations to the senior author is exaggerated (roughly one third of the total references).

Reviewer #4 (Remarks to the Author):

Report on manuscript NCOMMS-23-11343 The transition to synchronization of networked systems by A. Bayani et al.

In the present manuscript, the authors report on an analysis of the route to synchronization in diffusively coupled dynamical systems. The coupling is described by a Laplacian matrix. By virtue of the master stability function (MSF) formalism, they investigate the sequence of clusters whose dynamics join the synchronization manifold.

After a short, but sufficiently detailed recap of the MSF formalism, they introduce a new methodology to analyze the set of eigenvectors of the Laplacian matrix. This is illustrated by a toy example of a 10-node all-to-all network and further applied to synthetic networks and the case of the US power grid. In the numerical simulations, the nodal dynamics is assumed to be the Rossler or Lorenz system.

In general, I am impressed with the line of thought that provides an intuitive explanation of the emergence of complete synchronization; even more so as the topic of synchrony in dynamical systems has been extensively studied in the past.

To further improve the manuscript and make a strong case, I have the following questions, comments, and suggestions:

1. How can the authors be sure that the region of stability has at most a single interval on the ν -axis in Fig.1? See, for instance, A. Keane et al. Synchronisation in networks of delay-coupled type-I excitable systems, *The European Physical Journal B* 85, 407 (2012).
2. Why do the authors restrict their discussion to Laplacian matrices? There are other matrices with zero row sum, too.
3. To illustrate the sequence of synchronization, the author could provide a time series for increasing (ramping) coupling strength d , which would nicely supplement Fig.3.
4. On p.2, the authors mention periodic dynamics, but later consider only chaotic systems. Maybe, a demonstration for a normal form would increase the width of the applicability?
5. Speaking chaotic systems, the authors should discuss the effect of bubbling, that is, local regions of the synchronization manifold that have positive transversal Lyapunov exponents, but an intermittent return to the manifold.

6. The synthetically generated networks should be introduced by more than two references to papers (one of which is a preprint), where a subset of the authors are involved. I suggest to give a systematic investigation of standard network classes (random, scale-free, regular, small-world...). I understand that this would trigger a considerable amount of work, but the study would gain much in terms of its impact.

7. Why did the authors consider the US power grid? There are a number of other data sets. Their choice is not motivated.

8. Connected to the previous point, the combination of Rossler dynamics on a power-grid network appears strange and not very relevant. I would have expected a Kuramoto model with inertia, at the least.

Minor comments:

(a) double check hyphenation, for instance, in:

i. large scale simulation

ii. non identical

(b) p.3: The eigenvalues of the Laplacian matrix are characterized as “strictly real and positive”. First of all, the word “strictly” does not make sense. Secondly, there is one eigenvalue that is exactly zero, as correctly stated in the manuscript at a later stage. Therefore, the wording should be “real and non-negative” and could specify $0 = \lambda_1 \dots$

(c) The authors should refrain from using colloquial language such as:

“All needed ingredients are on the table, and one can now cook the cake!”

(d) In light of the analogue of an error function, it would make sense to use a factor of $1/2$ in the formula of E_{λ_i} . Then, the entries in Fig.2 with scale to the range $[0,1]$.

(e) The wording is awkward in parts:

i. order parameter -> control parameter

ii. theoretically

iii. demonstrate how high is

iv. One has that L is a symmetric, zero row sum, matrix. As so, it is diagonalizable...

v. our Supplementary Information text,

vi. both calculated along the synchronization solution's trajectory.

vii. p.4 orthogonal \rightarrow orthonormal

viii. ij entry vs. (ij) entry

(f) The full account of the considered systems should be given in terms of the parameters.

To sum up, I can see that the presented approach to synchronization might have its merits and might warrant publication in Nature Communications. For this, however, the study needs to gain in size by considering more general networks and models of the local dynamics.

REVIEWER COMMENTS

Reviewer #1 (Remarks to the Author)

Referee: This paper is concerned with a numerical method to accurately predict the sequence of events (emergence of synchronized clusters) that take place in a networked system during its transition to global (non-explosive) synchronization.

Answer: We would like to express our deep gratitude to this Referee, for all comments they made on our manuscript.

All comments, indeed, were extremely valuable to us, as they raised important issues, which, we agree, were not properly explained in the previous version of the manuscript. We are, therefore, grateful for the opportunity to clarify them, both here and in our revised submission.

We have, therefore, carried out a thorough revision of the presentation of our results and, in particular, of the points raised by this Referee.

We are pleased to present our detailed response/revisions below, point-by-point.

Referee: The first main drawback of this paper is that the Master Stability Function approach is based on the assumption of infinitesimal perturbation about a synchronous solution. Therefore, the stability of any cluster synchronization pattern depends on the related quotient network and on its dynamics, as shown in many works by Pecora et al. (e.g., Sorrentino, F., Pecora, L. M., Hagerstrom, A. M., Murphy, T. E., & Roy, R. (2016). Complete characterization of the stability of cluster synchronization in complex dynamical networks. *Science Advances*, 2(4), e1501737 and Siddique, A. B., Pecora, L., Hart, J. D., & Sorrentino, F. (2018). Symmetry and input-cluster synchronization in networks. *Physical Review E*, 97(4), 042217). This seems to be completely neglected by the authors.

Answer: The Referee is right when saying that the Master Stability Function (MSF) approach is a local approach, and as so it gives only a necessary, but not sufficient, condition for the stability of the synchronous solution. Of course, one can complement such an approach with other more “global” ones (*i.e.*, based on Lyapunov functionals), but then one immediately loses generality, as these approaches are possible only in a rather limited number of cases, whereas the MSF can be applied for any dynamical system. We agree with the Referee, however, that this point should have been made transparent to the reader, and therefore we added the needed clarification in Section 2 of the Main Text.

With respect to the Referee’s other point, the articles quoted by the Referee (and other similar previous work in the cluster synchronization literature) deal indeed with the problem of identifying groups of nodes (clusters) that can synchronize, and whether a cluster synchronized state is stable. This is expressed in terms of a partition of the nodes into orbits (for the whole, or a subgroup, of the automorphism group of the graph) or, more generally, for an equitable partition (respectively an external equitable partition, for Laplacian, rather than adjacency, dynamics).

However, the fact that a given (external) equitable partition can sustain cluster synchronization of its cells is different from *the problem of determining whether it will indeed be realized for some range of the coupling strength parameter and, crucially, in which order do clusters synchronize as this coupling increases from zero (which is what our study does).*

The knowledge that a graph admits a certain equitable partition (which is in general not unique), even if it is the (unique) coarsest equitable partition, *doesn't reveal how the cluster synchronization transition behaves as the coupling parameter increases*, which is the natural scenario investigated in our article.

For instance, one can have the coarsest equitable partition of a graph into say cells (clusters) C1, C2, and C3. A potential synchronization transition, as the coupling strength parameter increases, could be a subset of C1, followed by C3, followed by the whole of C1, followed by C2, followed by the whole network. Yes, the partition C1, C2, and C3 is eventually realized (just before full network synchronization), but that does not fully explain the complete synchronization transition path.

Not even the knowledge of all equitable partitions will guarantee that a given cell (cluster) will synchronize on its own: continuing with the same example, C2 might have an equitable subset which never synchronizes on its own, but only as part of C2, while that was not the case for the subset of C1 mentioned before (this example is not just theoretical speculation; see the new toy example we have added in Figure 3 in the Supplementary Information for a concrete example).

In our article, in contrast, we propose a method (the S_n algorithm) that *determines the synchronized clusters and the order in which they appear*, thus building on and improving previous results in the cluster synchronization literature.

Having said this, we fully recognize that we could have explained better how our work fits into the existing literature in our previous version of the Manuscript, and we apologize for that. We have now added a new paragraph in the introduction, better explaining how our work fits in the previous literature in cluster synchronization, including the references suggested by the Referee. We have also extended Section 2 in the Supplementary Information to clarify the relation between equitable partitions, symmetric orbits, and the clusters identified by our method (new Section 2.2 in the Supplementary Information). In particular, note that being a cluster in an (external) equitable partition is not enough to guarantee synchronization, not even a symmetric orbit — this is now illustrated in the new toy example in Figure 3 of the Supplementary Information, and represented schematically in the new diagram in Figure 1 of the Supplementary Information.

Referee: Further weaknesses:

- The network model is not so general: homogeneous, undirected, and Laplacian network, absence of delays, output function depending only on x_j . The class of considered networks should be better evidenced from the beginning of the paper, maybe also in its title.

Answer: Here, we have to respectfully disagree with the Referee. Indeed, we do not consider delays in our set-up (otherwise, a synchronization solution is not guaranteed), but with respect to the other criticisms of the generality of the network mode, we would like to note the following.

Homogeneous: It is true that the MSF approach assesses the stability of the synchronous solution, which formally exists only for identical systems. However, in our Figure 5b we give evidence that the same predicted sequence of clusters also characterizes the transition in heterogeneous networks, even for a rather large heterogeneity (fluctuation of about 10% in a parameter of the Rössler system). Moreover, it has been explicitly shown that the MSF formalism can be easily extended to networks of nearly non-identical systems [see, for instance, *Europhysics Letters* **85** (6), 60011 (2009)].

Undirected: Once again, we respectfully notice that the MSF approach has a vast literature, where it was extended to directed, or asymmetric, networks [see, for instance, Physical Review Letters **94**, 218701 (2005), Physical Review Letters **94**, 138701 (2005), and Phys. Rev. E **71**, 016116 (2005)]; and even - in specific cases- to non-diagonalizable networks [see Physical Review E **73**, 065106 (2006)]. Our study is pertinent, therefore, to all circumstances where the MSF can be applied, and we cannot say that it is limited to undirected graphs.

Laplacian and output function only depending on \mathbf{x}_j : As shown in Nature Communications **12** (1), 1255 (2021) – as well as in many other previous studies- the coupling term in Eq. (1) of the Main Text can be easily re-casted into $d \sum_{j=1}^N A_{ij} g(\mathbf{x}_i, \mathbf{x}_j)$, where A_{ij} are the elements of the adjacency matrix and $g(\mathbf{x}_i, \mathbf{x}_j)$ is a generic function (not necessarily diffusive) of both nodes' state vectors \mathbf{x}_i , and \mathbf{x}_j . The only requirement is that the coupling function g be “synchronization non-invasive” *i.e.*, $g(\mathbf{x}, \mathbf{x})=0$ for all \mathbf{x} . Then, when deriving the linear stability conditions, the Laplacian formalism comes out automatically as the natural consequence of being zero the total derivative of g calculated in the synchronization manifold. The derivation is rather cumbersome, but it is absolutely rigorous even for the case of higher-order interactions (*i.e.*, beyond the pairwise interaction limit, see again the mentioned Nature Communication paper). Now, this extends the validity of the MSF approach also to non-diffusive coupling functions (such as, for instance, $g(x,y)=\sin(x-y)$, or $g(x,y)=x^2y-xy^2$) and to coupling forms that do not use the Laplacian matrix.

Having said the above, we fully agree with the Referee that the class of considered networks (and its generality) should be clearly specified in the Main Text, which we have now corrected.

In fact, to avoid adding a large number of references about the various extensions of the MSF approach to the cases reported above (or to explicitly assume in our Eq.(1) a different coupling form which would then force us to introduce in the text a series of rather technical passages), we decided, as a compromise, to write an explicit long statement after Eq.(1) saying that our approach is valid under much more general cases, and that the only assumptions to be made are: *a*) a diagonalizable structure of connections, with a set of eigenvectors providing an orthonormal basis of \mathbb{R}^N , and *b*) synchronization non-invasiveness of the coupling functions, and to refer to a rather recent report on synchronization where the various extensions of the MSF are mentioned. We hope this is acceptable.

Referee: • The proposed results are concerned with networks with unrealistic dynamics, so there is no benchmark for the proposed results. Referring to (and making comparisons with) known results about real networks would make the paper much stronger. Talking about a “real-world network” for a network with real topology but unreal dynamics is misleading, in my opinion.

Answer: We are very sorry that the lack of clarity in our presentation was so significant that it led the Referee to raise as a weakness what we believe is a strength of our study.

A key point, indeed, of our study, is that the set of clusters that are synchronizing during the transition, the nodes that compose them, and the order in which clusters synchronize, are ***completely independent*** on the specific dynamical system that is controlling the evolution of the network's nodes (as they only depend on the spectrum of the Laplacian matrix, and consequently all these features only depend on the topology of the graph).

For instance, in Figure 3 of the Main Text, it is shown that the transition to synchronization is identical for the Rössler and Lorenz systems, with the only difference being the different critical values of the ν parameter at which the MSFs of these two systems cross the horizontal axis (0.179 for Rössler and 7.322 for Lorenz). In other words, the horizontal axes in Fig. 3a and 3b can be perfectly mapped one into the other by just multiplying for the ratio of the two critical values of the MSF.

This is, in our opinion, indeed a key strength of our study: it implies that in all practical situations (even when the knowledge of the dynamics of the nodes is fully unavailable, or when there are uncertainties in the model parameters) the entire sequence to synchronization can still be fully predicted.

We are nevertheless grateful to this Referee for having raised this comment as it prompted us to give the due emphasis to this point (by adding a sentence in the text, and by even modifying our abstract), which evidently was not apparent in the previous version of the Manuscript.

Referee: • A similar approach was already proposed in the following paper, not even cited by the authors: Fu, C., Lin, W., Huang, L., & Wang, X. (2014). Synchronization transition in networked chaotic oscillators: The viewpoint from partial synchronization. *Physical Review E*, 89(5), 052908.

Answer: We thank the Referee for pointing out this paper. The general method (eigenvalue analysis) is well known and obviously related to our approach in the sense that essentially any approach to synchronization stability analysis involves in one way or another a linearization around a stable state and a Lyapunov exponent analysis based on the interaction matrix (*e.g.*, Laplacian) eigenvalues. Our work goes beyond the results in Fu *et al.* Their approach, indeed, focuses on symmetry-induced partial synchronization (ours include this plus equitable cluster synchronization, and the study of the relation between cluster synchronization, symmetric orbits, and equitable partitions), considers only two toy unweighted models with 6, respectively 5, nodes (we provide toy, and real-world examples), the results are illustrated by, and extrapolated from, these examples, without, as far as we can see, general theoretical results (we provide theoretical guarantees relating spectral blocks, cluster synchronization, and the S_n algorithm, namely in Theorem 2.2 in the Supplementary Information).

Having said this, we agree with the Referee that, given the relevance of their work, we should have added it to the literature review, which we have done now.

Note that, although both the reference paper and our algorithm consider symmetries related to clusters during the transition, in the reference paper, the symmetries must be known or computed in advanced. In our algorithm, however, there is no need to know the symmetries, but only need the Laplacian matrix, and we are able to directly calculate all possible clusters appearing in the transition.

Furthermore, the reference approach focuses on the graph's symmetry orbits, but ours detects the intermediate structured states emerging in the path to synchrony of a network. These states are more general than the graph's symmetry orbits, with are (some, but not all) graph's symmetry orbits plus (some, but not all) equitable clusters.

Furthermore, the reference approach focuses on the graph's symmetry orbits, but ours detects the intermediate structured states emerging in the path to synchrony of a network. These states are more general than the graph's symmetry orbits, which are graph's symmetry orbits plus equitable clusters.

Referee: • The plots shown in Fig. 1 are correct if one considers global synchronization in Laplacian networks. But for cluster synchronization, each graph should change according to the considered synchronous clusters and to the network topology. How is this accounted for in the proposed method? This is related to the general drawback pointed out above.

Answer: We respectfully argue that this point is not quite correct. For instance, it was shown in Ref. [L. M. Pecora, F. Sorrentino, A. M. Hagerstrom, T. E. Murphy, and R. Roy, *Nature Communications* **5**, 4079 (2014)] that the stability properties of the synchronization solution limited to the nodes of a cluster (in that case, a symmetry orbit of the network) are determined by a proper subset of eigenvalues and eigenvectors of the Laplacian matrix of the entire network. In particular, if one considers a solution where only the nodes of a given cluster are synchronized, it was proven there that the space orthogonal to that solution has, as generating basis, a given subset of the original Laplacian eigenvectors. That reference is quoted in our Manuscript and was the very first to study the stability of cluster synchronization solutions.

Therefore, one can refer to a unique MSF (that calculated for the full network's synchronous solution, and reported in our Figure 1), and then read there directly the maximum Lyapunov exponents for each of the eigenvectors transverse to the cluster solution (*i.e.*, for each of the eigenvectors forming the proper subset that identifies that cluster).

We could have included the rather lengthy calculations from Pecora *et al.* that demonstrate this point, for clarity, but, in order not to overly burden the presentation, we have added a sentence that refers the reader to the mentioned Manuscript for more details.

Referee: • In all the proposed examples the node dynamics are chaotic, even if the literature numbers many examples of analysis of networks whose node dynamics are periodic. At least one example of this kind would be useful, mainly in the presence of intertwined clusters. In any case, additional evidence is needed to support the main claims of the paper.

Answer: Indeed, we made a purposeful choice to focus on chaotic systems, where the synchronization transition is not trivial. In fact, considering periodic (identical) systems implies the following issues. In the MSF, $v = 0$ corresponds to $\lambda_1 = 0$ *i.e.*, to the eigenvector \mathbf{v}_1 aligned with \mathcal{M} . Therefore, $\Lambda(0)$ is equal to the maximum Lyapunov exponent of the isolated system $\dot{\mathbf{x}} = \mathbf{f}(\mathbf{x})$ which, for a periodic system, is equal to 0. In Class II, one has then that the critical value of v^* vanishes, as well as the threshold for complete synchronization. Therefore, no intermediate clusters can be observed and the entire network synchronizes for any infinitesimal coupling strength (as it is obvious). In Class III, we have $v_1^* = 0$ and $v_2^* \neq 0$. Therefore, the transition to synchronization will be exactly as the one of Class II (*i.e.*, with no clusters, and with a setting of the complete network's synchronization state at infinitesimal values of the coupling strength), and the de-synchronization transition, instead, will be through a sequence of clustered states. Now, the desynchronization features are not the scope of this paper (we do, indeed, have a complete description also of this transition in Class III, but it will be published elsewhere, as there are significant conceptual differences). This should hopefully clarify why we decided, from the start, to concentrate on chaotic systems, as in this case the transition to synchronization is not trivial.

Having said this, we fully agree with the Referee that additional evidence was needed in support of our main claims, and consequently we have added a new Figure (Figure 6) reporting the application

of our method to two other real-world networks: the Yeast protein-protein interaction network and the ego-Facebook network. Furthermore, we also considered two further applications to synthetic networks in the Supplementary Information for a better illustration of the nature of the observed clusters. In this way, indeed, evidence is given that our method finds effective applications for all kinds of real-world networks: for man-made technological networks (as the US Power Grid), for social networks (as the ego-Facebook network) and for biological networks as well (as the Yeast interaction networks). We are hopeful that the Referee will agree that this is sufficient evidence in support of the main claims of our paper.

Referee: • The section titled “The synchronized solution” contains only known results and could be moved to the Supplementary Information.

Answer: We agree that this section contains the derivation of the MSF (a somehow well-established result), but, unfortunately, the interpretation that was given to the MSF results has not always been correct in the cluster synchronization literature.

For instance, instead of focusing on the classification reported in our section, a huge literature exists which gave emphasis to the concept of “synchronizability” and compared different network’s topologies in terms of the ratio $r = \lambda_N / \lambda_2$ between the maximum and the second smallest eigenvalues of the Laplacian matrix, arguing that the closest to 1 is that ratio, the more synchronizable is the network. Unfortunately, this concept (without a proper and careful reference to the classification reported in our section) may bring in most cases to several meaningless (or even wrong) conclusions.

For instance, in Class I systems, synchronization is clearly not achievable. Therefore, it does not make sense to classify networks for this type of systems, in terms of their “synchronizability”, or to measure the eigenvalue ratio, as no network will synchronize for this kind of systems.

In Class II, the range of coupling strength for which synchronization is stable is infinite, and the threshold is only controlled by λ_2 , as we carefully argue in our section. Therefore, again, classifying networks by means of the eigenvalue ratio does not make sense. To give a concrete example, consider two networks G_1 and G_2 , and suppose that one has $\lambda_2(G_1) = 1$ and $\lambda_N(G_1) = 2$, as well as $\lambda_2(G_2) = 10^{45}$ and $\lambda_N(G_2) = 10^{46}$. Now, according to the synchronizability criterion reported in that literature, one should conclude that G_1 (for which $r=2$) is more synchronizable than G_2 (for which $r=10$), while the threshold for synchronization of G_2 is 45 (!) orders of magnitude smaller than that of G_1 (*i.e.*, more orders of magnitude than those existing between the smallest and the largest scale in our universe).

In Class III, the ratio r may have a (arguably not so important) meaning only to compare the range of coupling strengths for which networks may sustain a synchronous solution, but once again the threshold is only determined by λ_2 , and the above counterexample still holds.

We find it unfortunate that such a large attention has been given to this parameter, without explicitly linking it to the classification scheme as explained in the section. Therefore, we think that the classification of dynamical systems and output functions emerging from the MSF is a sufficient important point, which deserves to be clarified once and for all, and that’s why we feel necessary to maintain the section within the body of the Main text.

Referee: • Condition 1 in the proposed Theorem is claimed to be “more specific than the graph’s equitable partitions”, but in several other papers the same condition is already used to find equitable clusters, e.g.

o [13] M. T. Schaub, N. O’Clery, Y. N. Billeh, J.-C. Delvenne, R. Lambiotte, and M. Barahona, “Graph partitions and cluster synchronization in networks of oscillators,” *Chaos: An Interdisciplinary Journal of Nonlinear Science*, vol. 26, no. 9, p. 094821, 2016.

o [14] A. B. Siddique, L. Pecora, J. D. Hart, and F. Sorrentino, “Symmetry-and input-cluster synchronization in networks,” *Physical Review E*, vol. 97, no. 4, p. 042217, 2018.

Answer: We thank the Referee for this comment and recognize that we did not explain very well the relation between equitable partitions, symmetric orbits, and the clusters in the Theorem. It is true that our notion of cluster synchronization is more specific than an equitable partition: an equitable partition does *not* guarantee that each cluster satisfies the condition in the theorem, which guarantees cluster synchronization at, and from, a given critical value of the coupling strength parameter. This is also true for symmetric orbits (orbits of the automorphism group of the graph/network), which may not satisfy the conditions in the Theorem. We have now coined the term Strong Equitable Cluster (SEC) to define the clusters where the Theorem in the Main Text (equivalently, Theorem 2.2 in the Supplementary Information) applies, and clarified the relation to equitable partition and symmetric orbit in the new Section 2.s of the Supplementary Information. We have also extended the concept of SEC to relative SEC and explained the relation between equitable partition and relative SECs.

Namely, we define a Strong (External) Equitable Cluster (SEC) as a subset of nodes such that the connectivity with every other node in the network is independent of the chosen node in the cluster (this is Condition 1 in Theorem 2.2. in the Supplementary Information). This is related, but not the same, to the notion of a cell in an (external) equitable partition: a subset of nodes C is a SEC if and only if the partition of the nodes into C and clusters of size one (one per node) is equitable. Equivalently, a SEC is a subset of nodes C where every other node in the network is connected to all, or none, of the nodes in C . Our Theorem (Theorem 2.2 in the Supplementary Information) guarantees that C synchronizes when its critical value (which we show equals the smallest nonzero Laplacian eigenvalue of the graph induced by C plus its external degree, that is, the number of nodes outside C connected to one, and hence all, the nodes of C). This is not surprising: every node in C receives the same input from each node not in C .

As summarized schematically in Figure 1 of the Supplementary Information (new), not every cell in an equitable partition (we call it an Equitable Cell, EC) is a SEC, not even an orbit of the automorphism group (which we call a Symmetric Orbit, SO).

We complement the notion of SEC with a more general notion of relative SEC with respect to clusters C_1 to C_r . This means, by definition, that the sum of the connections of a node of the relative SEC, call it C , to all the nodes in C_i is independent of the chosen node in C . This is not the same as an equitable partition, which also requires equitability (in the obvious sense) between C_i and C_j for all i not equal to j . If each of the clusters C_1 to C_r are synchronized (a big ‘if’), then so is the relative SEC C , as, again, every node in C receives the same input. Note that all our examples (found through the proposed S_n algorithm), all the cluster synchronization examples correspond to SECs, or in SECs relative to already synchronized, or simultaneously synchronized, clusters. (This situation is nicely illustrated in the new toy example shown in Figure 3 in the Supplementary Information.)

A natural question, which we believe the Referee is getting at, is to explain how this fits in with previous results, such as the ones cited by the Referee, regarding cluster synchronization in equitable partitions. First, note that we are investigating a related but different question: previous results in the literature, including the references provided by the Referee, identify clusters where synchronization is possible (for instance, by setting equal initial conditions on each cell of an equitable partition); this is different from identifying the actual cluster synchronization transitions (which clusters, in which order) as the coupling parameter increases.

Some equitable partitions, although theoretically possible, are never realized, unless we start with an already synchronized state. All in all, our problem is not to identify partitions/clusters supporting cluster synchronization, but realizable cluster synchronization as we increase the coupling strength parameter. Just to briefly illustrate this point, if C_1 to C_r are the cells of an equitable partition, each cell is a SEC relative to the other cells, which guarantees synchronization when the critical values of all the cells are reached, but not necessarily before. In particular, it doesn't explain the order at which the cluster synchronization occurs, nor partial synchronization of, nor merger between, subsets of the equitable cells, as detected by the S_n algorithm put forward in our article.

Having said this, the primary purpose of our article is to validate the S_n algorithm's ability to exactly find the cluster synchronization transitions. The correspondence to certain structural subsets of nodes (SECs and relative SECs) is an added contribution. A separate question, beyond the scope of our paper, is to find an algorithm to uncover the hierarchical structure of SECs and relative SECs in synchronization order purely from the graph/network structure, their critical coupling constant (at which the corresponding cluster synchronization occurs) and to mathematically prove that cluster synchronization can only occur in this way.

Referee: • There are some typos, e.g., “cluterize”, “The only different will be”.

Answer: Thank you for picking this up, we have now corrected them.

In summary, we are very grateful to this Referee for the constructive and insightful comments, that prompted us to improve our results, their generality and the clarity of their presentation. We have now carefully considered all the Referee's concerns and made the necessary revisions in our Manuscript. On the light of the answers provided above and of the modifications made to our presentation, we hope that the Referee will now find our Manuscript suitable for publication.

----- o -----

Reviewer #2 (Remarks to the Author)

Referee: In this paper the authors study a rather general class of coupled dynamical systems operating on the nodes of complex networks. The dynamical process involves both an on-site dynamics (that can be possibly be chaotic) as well as a pairwise coupling between nodes, mediated by the graph Laplacian matrix.

The focus is on analyzing the progressive synchronization process, which emerges as an overall coupling-strength parameter, " d ", is increased. For this, the authors rely on the well-known formalism of the Master Stability function (MSF). This allows them to separate the diverse "modes" associated with each of the eigenvalues of the Laplacian matrix and study their stability independently from each other. As d is increased a series of local synchronization events progressively unfolds following the sequence of Laplacian modes that become sequentially stabilized (i.e. their associated eigenvalues cross the stability limit). Eventually, the completely synchronized state becomes stabilized.

I would say that such a setup ---allowing to describe in terms of features of the Laplacian graph the cascade or partial synchronization events leading eventually to global synchronization--- is very well known and well established in the literature.

The main novelty of the present manuscript is that it considers the specific case of networks with certain symmetries, i.e., assuming the existence of groups of nodes that can be permuted between themselves without affecting their effect on the rest of the network, or in other words, they are indistinguishable nodes (to the "eyes of any other node of the network", to use the author's wording). Such groups of indistinguishable nodes are called "clusters". The main contribution of the manuscript is proving mathematical theorem, by virtue of which if a cluster has k nodes, then there is a spectral block defined by $k-1$ Laplacian eigenmodes, perfectly localized at exactly those k nodes.

A direct consequence of the theorem is that, for networks with the aforementioned clusters, the sequence of eigenvalues crossing the stability limit as " d " is increased, can be associated with an easily interpretable description of the cascade of synchronization events. As a matter of fact, as the corresponding eigenvectors can be localized in clusters, the synchronization unfolds by progressively generating local synchronous clusters and then merging them.

Answer: We would like to thank this Referee for the concise summary they did of our work, and for having clearly summarized the novel aspects of our paper.

To the best of our knowledge, our approach to the problem of identifying the exact sequence of clusters that characterize the transition to synchronization is novel. The fact that we could give rigor to our work by proving a theorem that explains why local synchronous clusters are generated, and the fact that this way one can demonstrate universality of the transition (i.e., the fact that the sequence of such clusters is totally independent on the specific dynamical system ruling the evolution of the network's nodes) give, in our opinion, strength and relevance to our results.

With this in mind, we found the Referee's concerns below truly insightful, and we are happy to present our detailed response/revisions below, point-by-point.

Referee: The authors design a simple and clear protocol to actually do this and identify the synchronizing clusters in a progressive way and apply it to a simple toy network with some symmetries. The algorithm is verified to work fine both for small and for large networks provided that they have clusters (otherwise, there is no prediction to be made).

Finally, the method is also applied to a real example: the US power grid network. The authors detect a well-defined series of synchronization events and clusters and analyze how robust is the synchronization cascade when some degree of heterogeneity is added.

My main concern is about the actual relevance of the findings for real-world applications/networks. As I already wrote, the description of the unfolding of synchronization exploiting the properties of the Laplacian matrix is already well known, so the really new contribution of this work is that of extending those old ideas to networks with clusters. Thus, my feeling is that the authors should make a much stronger case justifying that clusters are generically present in real-world networks and show some other example (e.g. brain networks?) complementing the power grid one, to really convince the reader of the power of identifying clusters and applying the here-developed tools to uncover the route to synchronization in real-world networks.

Answer: This is a very relevant point, and it is now clear that we had not explained it very well in the previous version of our Manuscript.

In fact, clusters due to symmetries are ubiquitous, as most real-world networks have a large number of localized symmetries, that is, a large number of small subgraphs called symmetric motifs [MacArthur2008, Sanchez2020], where independent (formally, support-disjoint) symmetries are generated, with each symmetric motif made of one or more orbits. For example, in [Sanchez2020], the real-world test networks display a large number of symmetric motifs (SMs), relative to the size (number of nodes n) of the networks: from 149 SMs (protein-protein interaction network of the yeast, $n=1647$) to over 245k (LiveJournal social network, $n>5m$). Indeed, most networks are highly symmetric, with up to 60% of the nodes involved in some non-trivial symmetries (that is, belonging to a symmetric motif). In turn, most symmetric motifs in real-world networks have one orbit (over 90% in all examples in [Sanchez2020], for instance). More examples of symmetries in real-world symmetries can be found in e.g., [MacArthur2009, Ball2018, Ward2021].

[MacArthur2008] MacArthur, Ben D; Sánchez-García, Rubén J; Anderson, James W. Symmetry in complex networks. *Discrete Applied Mathematics* **156** 18 3525-3531 (2008).

[MacArthur2009] MacArthur, Ben D; Sánchez-García, Rubén J; Spectral characteristics of network redundancy. *Physical Review E* **80** 2 26117 (2009).

[Ball2018] Ball, Fabian, and Andreas Geyer-Schulz. "How symmetric are real-world graphs? A large-scale study." *Symmetry* **10** 29 (2018).

[Sanchez2020] Sánchez-García, Rubén J. Exploiting symmetry in network analysis. *Communications Physics* **3** 1 87 (2020).

[Ward2021] Ward, J. A. Dimension-reduction of dynamics on real-world networks with symmetry. *Proceedings of the Royal Society A*, **477** 2251 20210026 (2021).

We apologize we didn't emphasize the relevance of symmetries in real-world networks well enough in our previous version of the Manuscript; we have now included a better explanation in the introduction, with some of the references above in the Main Text.

Having said this, note that our method goes beyond symmetric orbits. The synchronized clusters uncovered by our method are different from the symmetric orbits or even the equitable partitions typically studied in previous work in cluster synchronization. Namely, we find, through our S_n algorithm based on the unfolding of synchronization given by the Laplacian eigenvectors, that cluster synchronization occurs on what we now call Strongly Equitable Clusters (SECs). These are the clusters satisfying the hypothesis of the Theorem in the Main Text (Theorem 2.2 in the Supplementary Information). In the new version of our article, we have clarified the relation between SECs, equitable partitions, and symmetric orbits. Note that it is a more restrictive concept: not every cluster in an equitable partition, not even the orbit partition of the automorphism group of the graph, is necessarily a SEC (see Figure 1 in the new Section 2.2 of the Supplementary Information). Crucially, though, symmetric motifs with one orbit (over 90% of them in a typical real-world network) are necessarily a SEC, so these provide clusters with are guaranteed to synchronize, independently from the rest of the network, when their critical threshold (the value of the smallest non-zero Laplacian eigenvalue plus the external degree of the subgraph) is reached by the coupling strength parameter. (Note that symmetric motifs with one orbit include the clusters characterized in Theorem 2.3 in the Supplementary Information.)

All in all, we agree with the Referee that the presence of symmetries in real-world networks justifies the applicability of our algorithm in the study of synchronization in real-world systems, and this is indeed the case, as explained above.

We have added a discussion at the end of Section 2, namely a new Section 2.2 in the Supplementary Information to define SEC and clarify the relation to symmetric orbits and equitable partitions, adding a schematic figure (Figure 1) and two toy examples (Figures 2 and 3) to further illustrate.

Furthermore, we have added a new Figure (Figure 6) reporting the application of our method to two other real-world networks: the Yeast protein-protein interaction network and the ego-Facebook network. In this way, evidence is given that our method finds effective applications for all kinds of real-world networks: from man-made technological networks (as the US Power Grid) to social networks (as the ego-Facebook network) or biological networks as well (as the Yeast interaction networks).

Referee: In summary, I think that the paper is clear, sound and well-written. It certainly introduces new ideas and should be published in some form or the other. But depending on how strongly can the authors defend the importance of clusters for real-world networks. I would either recommend accepting the paper for Nat. Comm. or either suggest it to be considered (accepted) in a less generalistic venue such as Communications Physics.

Answer: We thank this Referee for the positive feedback and hope we have addressed this issue fully in our response above, in terms of both the importance and ubiquity of symmetries in real-world networks.

Referee: Minor comments:

- I believe there is a mistake/misprint in the first network of Fig.2. as entry (7,10) does not coincide with (10,7).

Answer: The Referee is absolutely right: the matrices S_n are symmetric and there was a misprint in the first network of Figure 2, which is now corrected.

- I could not find the values of the weights in the discussed example. I think it is important to include them so that the reader can reproduce the calculation in detail. Moreover, it is important to further underline that these weights do not need to take the same values within a cluster. E.g. in the discussed example it seems to me that nodes 8 and 9 are more strongly connected between themselves than e.g. 7 and 10, even if they all are within the same cluster.

Answer: Indeed, the definition of synchronizable cluster in our theorem **does not imply any condition** on the way nodes within the cluster are connected **among them**. Therefore, it may well be that the connections between nodes belonging to a same cluster have different weights.

However, we agree we should have given full information on the weights of the example in Figure 2, so as a reader can reproduce all details of our calculations. Therefore, we have now reported the adjacency matrix of the network of Figure 2 in the Supplementary Information.

We also recognize that we did not explain very well the relation between equitable partitions, symmetric orbits, and the clusters in the Theorem. We have now added a new Section in the Supplementary Information, where the relation between the clusters of our theorem and the equitable partitions and symmetric orbits is fully clarified.

- I would like the authors to discuss what happens if the dynamical process does not explicitly involve a Laplacian matrix for the coupling between nodes but, e.g., an adjacency one. In other words, the Laplacian already implies some conservation law that might be too restrictive for many real-world potential applications.

Answer: A similar point was raised by Referee n.1 and 4. We, therefore, give here a similar answer.

As shown in Nature Communications **12** (1), 1255 (2021) – as well as in many other previous studies– the coupling term in Eq. (1) of the main text can be easily re-casted into $d \sum_{j=1}^N A_{ij} g(\mathbf{x}_i, \mathbf{x}_j)$, where A_{ij} are the elements of the adjacency matrix and $g(\mathbf{x}_i, \mathbf{x}_j)$ is a generic function (not necessarily diffusive) of both nodes' state vectors \mathbf{x}_i , and \mathbf{x}_j . The only requirement is that the coupling function g be “synchronization non-invasive” *i.e.*, $g(\mathbf{x}, \mathbf{x}) = 0$ for all \mathbf{x} . Then, when deriving the linear stability conditions, the Laplacian formalism comes out automatically as the natural consequence of being zero the total derivative of g calculated in the synchronization manifold. The derivation is rather cumbersome, but it is absolutely rigorous even for the case of higher-order interactions (*i.e.*, beyond the pairwise interaction limit, see again the mentioned Nature Communication paper). Now, this extends the validity of the MSF approach also to non-diffusive coupling functions (such as, for instance, $g(x,y)=\sin(x-y)$, or $g(x,y)=x^2y-xy^2$) and to coupling forms that do not use the Laplacian matrix.

Having said the above, we fully agree with the Referee that the class of considered networks (and its generality) should be specified more clearly in the main text. Now, to avoid adding a large number of references about the various extensions of the MSF approach to the cases reported above (or, even worse, to explicitly assume in our Eq. (1) a different coupling form which would then forces us to introduce in the text a series of rather technical passages), we decided, as compromise, to write an explicit long statement after Eq. (1) explaining how our approach is valid under much more general cases, and that the only assumptions to be made are: *a*) a diagonalizable structure of connections, with a set of eigenvectors providing an orthonormal basis of \mathbb{R}^N , and *b*) synchronization non-invasiveness of the coupling functions, and to refer to a rather recent report on synchronization where the various extensions of the MSF are mentioned.

- Honestly, I think that the number of self-citations to the senior author is exaggerated (roughly one third of the total references).

Answer: Thank you for raising this issue. We have amended the reference list consequently. In particular, several new references of studies made by other groups have been added and three citations to the senior author have been removed.

In summary, we would like to thank this Referee for their positive evaluation of our work, as well as for the many pertinent and insightful comments, that helped us to improve our results, their generality, and the clarity of their presentation in this new version of our Manuscript. We have carefully considered all the Referee's concerns and made the necessary revisions in our Manuscript. We therefore hope that this Referee will now be inclined to accept our Manuscript for publication.

----- 0 -----

Reviewer #4 (Remarks to the Author):

Report on manuscript NCOMMS-23-11343 The transition to synchronization of networked systems by A. Bayani et al.

Referee: In the present manuscript, the authors report on an analysis of the route to synchronization in diffusively coupled dynamical systems. The coupling is described by a Laplacian matrix. By virtue of the master stability function (MSF) formalism, they investigate the sequence of clusters whose dynamics join the synchronization manifold.

After a short, but sufficiently detailed recap of the MSF formalism, they introduce a new methodology to analyze the set of eigenvectors of the Laplacian matrix. This is illustrated by a toy example of a 10-node all-to-all network and further applied to synthetic networks and the case of the US power grid. In the numerical simulations, the nodal dynamics is assumed to be the Rossler or Lorenz system.

In general, I am impressed with the line of thought that provides an intuitive explanation of the emergence of complete synchronization; even more so as the topic of synchrony in dynamical systems has been extensively studied in the past.

Answer: We wish to thank the Referee for this introduction. We agree with the broader context that the Referee offers in the comment for our contribution. For us too, the excitement around our work is rooted in the fact that we have arrived to a complete solution of a rather old problem (synchronization of networked systems) by trying to formulate a fresh novel approach which brings at the same time intuition and rigor into a one of the most well-known and empirically relevant phenomena of network science.

We are happy to see that this message, resonated by the Referee, indeed, came through.

Referee: To further improve the manuscript and make a strong case, I have the following questions, comments, and suggestions:

1. How can the authors be sure that the region of stability has at most a single interval on the ν -axis in Fig.1? See, for instance, A. Keane et al. Synchronisation in networks of delay-coupled type-I excitable systems, The European Physical Journal B 85, 407 (2012).

Answer: The Referee is completely right: the region of stability may be formed by the union of several intervals on the ν axis. However, the aim of our work is to describe the **first** transition to synchronization, *i.e.*, the process that starts at $d = 0$ and develops when progressively increasing the coupling strength. We have shown that a sequence of events occurs, and each of them can be predicted.

When a system will show multiple regions of stability in the ν axis the situation will be as follows.

First, one will observe the transition to synchronization, as described in our Manuscript.

Then, another de-synchronization transition will occur when the spectrum of eigenvalues will start leaving the first region of stability. Let us anticipate you that we also have a full description of this latter scenario, and that the sequence with which clusters de-synchronize has not a trivial correlation with the sequence (described in our paper) with which they synchronize. All the corresponding material will be presented by us elsewhere, since describing de-synchronization phenomena is out of the scope of the present submission.

When the de-synchronization sequence will be finalized (and the spectrum of eigenvalues will start entering the second region of stability), another sequence of cluster synchronization will happen **identically** to that described in our Manuscript.

In practice, various synchronization (de-synchronization) transitions (as many as the stability regions of the system in the ν axis) will occur, each of which identical to the first synchronization (de-synchronization) transition observed. As said, we have full results on this scenario, which we are preparing the submission elsewhere.

Referee: Why do the authors restrict their discussion to Laplacian matrices? There are other matrices with zero row sum, too.

Answer: A similar point was raised by Referees n.1 and 2. We, therefore, give here a similar answer.

As shown in Nature Communications **12** (1), 1255 (2022) – as well as in many other previous studies- the coupling term in Eq. (1) of the main text can be easily re-casted into $d \sum_{j=1}^N A_{ij} g(\mathbf{x}_i, \mathbf{x}_j)$, where A_{ij} are the elements of the adjacency matrix and $g(\mathbf{x}_i, \mathbf{x}_j)$ is a generic function (not necessarily diffusive) of both nodes' state vectors \mathbf{x}_i , and \mathbf{x}_j . The only requirement is that the coupling function g be “synchronization non-invasive” *i.e.*, $g(\mathbf{x}, \mathbf{x})=0$ for all \mathbf{x} . Then, when deriving the linear stability conditions, the Laplacian formalism comes out automatically as the natural consequence of being zero the total derivative of g calculated in the synchronization manifold. The derivation is rather cumbersome, but it is absolutely rigorous even for the case of higher-order interactions (*i.e.*, beyond the pairwise interaction limit, see again the mentioned Nature Communication paper). Now, this extends the validity of the MSF approach also to non-diffusive coupling functions (such as, for instance, $g(x,y)=\sin(x-y)$, or $g(x,y)=x^2y-xy^2$) and to coupling forms that do not use the Laplacian matrix.

Having said the above, we however fully agree with the Referee that the class of considered networks (and its generality) should be specified more clearly in the main text. Now, to avoid adding a large number of references about the various extensions of the MSF approach to the cases reported above (or, even worse, to explicitly assume in our Eq. (1) a different coupling form which would then forces us to introduce in the text a series of rather technical passages), we decided, as a compromise, to write an explicit long statement after Eq. (1) explaining how our approach is valid under much more general cases, and that the only assumptions to be made are: a) a diagonalizable structure of connections, with a set of eigenvectors providing an orthonormal basis of \mathbb{R}^N , and b) synchronization non-invasiveness of the coupling functions, and to refer to a rather recent report on synchronization where the various extensions of the MSF are mentioned.

Referee: To illustrate the sequence of synchronization, the author could provide a time series for increasing (ramping) coupling strength d , which would nicely supplement Fig.3.

Answer: Many thanks for this suggestion. Unfortunately, when we tried a plot showing a time series for increasing (ramping) values of d , it contained too many oscillations and the synchronization phenomena were not so clearly visible.

On the other hand, we fully agree with the suggestion of the Referee that Figure 3 could be nicely supplemented. Therefore, we have reported snapshots of the nodes' dynamics within each of the 4 dynamical regimes predicted (new panels b1-b4 of Figure 3), so as to give the reader a more visual illustration on the sequence of synchronization features observed during the transition.

Referee: On p.2, the authors mention periodic dynamics, but later consider only chaotic systems. Maybe, a demonstration for a normal form would increase the width of the applicability?

Answer: A similar point was raised by Referee n.1. We therefore give here a similar answer. Considering periodic (identical) systems implies the following issues.

In the MSF, $\nu = 0$ corresponds to $\lambda_1 = 0$ *i.e.*, to the eigenvector \mathbf{v}_1 aligned with \mathcal{M} . Therefore, $\Lambda(0)$ is equal to the maximum Lyapunov exponent of the isolated system $\dot{\mathbf{x}} = \mathbf{f}(\mathbf{x})$ which, for a periodic system, is equal to 0.

In Class II, one has then that the critical value of ν^* vanishes, as well as the threshold for complete synchronization. Therefore, no intermediate clusters can be observed and the entire network synchronize for any infinitesimal coupling strength (as it is obvious).

In Class III. One will have $\nu_1^* = 0$ and $\nu_2^* \neq 0$. Therefore, the transition to synchronization will be exactly as the one of Class II (*i.e.*, with no clusters, and with a setting of the complete network's synchronization state at infinitesimal values of the coupling strength), and the de-synchronization transition, instead, will be through a sequence of clustered states. Now, the desynchronization features are not the scope of this paper (we do, indeed, have a complete description also of this transition in Class III, but it will be published elsewhere, as there are significant conceptual differences).

The reasons summarized above made that we decided, since the beginning, to concentrate on chaotic systems, as there the transition to synchronization is not trivial.

Referee: Speaking chaotic systems, the authors should discuss the effect of bubbling, that is, local regions of the synchronization manifold that have positive transversal Lyapunov exponents, but an intermittent return to the manifold.

Answer: To the best of our knowledge, the bubbling phenomenon is an effect observed in certain nonlinear dynamical systems where a change in parameter results in a qualitative change in the way the attractor responds to noise and/or other dynamical perturbations. The qualitative manifestation of the bubbling transition is normally the emergence of intermittent bursts of chaotic trajectories away from a previously constrained attractor.

It is known that bubbling may affect the manifold of a completely synchronized solution.

As so (and considering the fact that, as shown in [Nature Communications 5, 4079 (2014)], the stability properties of the synchronization solution limited to the nodes of a cluster are entirely determined by a proper subset of eigenvalues and eigenvectors of the Laplacian matrix of the entire

network) one can reasonably expect that bubbling may also characterize synchronization of the clusters that emerge during the transition.

However, an in-depth study of the bubbling phenomenon would imply considering all different invariant subsets (such as the unstable periodic orbits) embedded within the manifold described by the synchronization solution, and checking if and when they lose stability in one of the transverse directions.

This, once again to the best of our knowledge, was never done before in the context of the Master Stability Function (i.e., for studying the bubbling bifurcation in the complete network synchronization state) and would imply an even more sophisticated (and different from cluster to cluster) treatment for each of the cluster synchronized solutions that are shown to appear in the transition.

We feel that the Referee's suggestion is very deep and valuable, and motivated by the Referee's point we have indeed started thinking about this problem (that, we confess, we had not considered before). This could indeed be a very interesting problem, for instance, a given unstable periodic orbit embedded in the chaotic attractor may lose stability in one direction transverse to the manifold corresponding to one of the clusters' solutions, while is stable in all directions transverse to the manifold corresponding to another cluster's solution, giving rise therefore to a sort of "selective" bubbling behavior, which may be observed in some clusters while is not observed in some others. This, we believe, is out of the scope of the present research, and deserves separate future investigation.

Referee: The synthetically generated networks should be introduced by more than two references to papers (one of which is a preprint), where a subset of the authors are involved. I suggest to give a systematic investigation of standard network classes (random, scale-free, regular, small-world...). I understand that this would trigger a considerable amount of work, but the study would gain much in terms of its impact.

Answer: We thank the Referee for having raised this point.

Our study demonstrates that the entire transition to synchronization is ruled (under the assumptions made, see the above answers) by the eigenvalues and eigenvectors of the Laplacian matrix. In this sense, the specific scaling of the degree distribution (i.e., whether a graph is scale-free or is endowed with exponentially decaying tails, like in random networks) is not essential. Nor is it essential whether the network's diameter scales linearly (as in regular graphs) or logarithmically (as in small-world networks) with the network's size. From this point of view, therefore, we are unsure whether the systematic investigation suggested by the Referee would lead to conclusive results.

We, however, fully agree with the Referee's point that our study would gain much more impact by providing more general (and more related to real-world examples) applications. Therefore, we decided to include a series of new illustrative applications. In particular,

- 1) We have added a discussion at the end of Section 2 (namely the new Section 2.2) in the Supplementary Information which clarifies the relation between the observed clusters and the graph's symmetric orbits and equitable partitions, and we added a schematic figure (Figure 1) and other two synthetic examples (Figures 2 and 3) for illustration.
- 2) Furthermore, we have added a new Figure (Figure 6) to the main text, reporting the application of our method to two other real-world networks: the Yeast protein-protein interaction network

and the ego-Facebook network. In this way, evidence is given that our method finds effective applications for all kinds of real-world networks: from man-made technological networks (as the originally examined US Power Grid), to social networks (as the ego-Facebook network) and biological networks as well (as the Yeast interaction networks).

We are confident, and hope the Referee will agree, that the evidence provided in support of the main claims of our paper is sufficiently justified.

Referee: Why did the authors consider the US power grid? There are a number of other data sets. Their choice is not motivated.

Answer: We fully agree with the Referee that on the need of providing more evidence of applicability. We therefore have added a new Figure (Figure 6) reporting the application of our method to two other real-world networks: a Yeast protein-protein interaction network, and an ego-Facebook network. This provides evidence that our method can be effectively applied in all kinds of real-world scenarios: for man-made technological networks (such as the US Power Grid), to social networks (such as the ego-Facebook network) or biological networks (such as the Yeast interaction network).

Referee: Connected to the previous point, the combination of Rössler dynamics on a power-grid network appears strange and not very relevant. I would have expected a Kuramoto model with inertia, at the least.

Answer: We are truly sorry that the lack of clarity in our presentation was so that it generated confusion.

A key point, indeed, of our study, is that the set of clusters that are synchronizing during the transition, the nodes that compose them, and the order in which clusters synchronize are *completely independent* on the specific dynamical system that is controlling the evolution of the network's nodes (as they only depend on the spectrum of the Laplacian matrix, and consequently all these features only depend on the topology of the graph). Therefore, it is independent of the specific dynamical model used to describe the single dynamical unit of the network: the complete sequence of clusters observed in the sequence, and their exact order, will be always predicted correctly.

For instance, in Figure 3 of the Main Text, it is shown that the transition to synchronization is identical for the Rössler and Lorenz systems, with the only difference being the different critical values of the ν parameter at which the MSFs of these two systems cross the horizontal axis (0.179 for Rössler and 7.322 for Lorenz). In other words, the horizontal axes in Fig. 3a and 3b can be perfectly mapped one into the other by just multiplying for the ratio of the two critical values of the MSF.

This, we believe, is a very strong result of our study because it implies that in all practical situations (even when the knowledge of the dynamics of the nodes is fully unavailable, or when there are uncertainties in the model parameters) the entire sequence to synchronization can still be fully predicted.

We are grateful to this Referee for having raised this comment because it definitely urged us to give the due emphasis to this point (by adding a sentence in the text, and by even modifying our abstract), which evidently was not apparent in the previous version of the Manuscript.

Referee: Minor comments:

(a) double check hyphenation, for instance, in: i. large scale simulation
ii. non identical

Answer: We have now corrected this, many thanks.

(b) p.3: The eigenvalues of the Laplacian matrix are characterized as “strictly real and positive”. First of all, the word “strictly” does not make sense. Secondly, there is one eigenvalue that is exactly zero, as correctly stated in the manuscript at a later stage. Therefore, the wording should be “real and non-negative” and could specify $0 = \lambda_1$

Answer: That’s absolutely right, apologies for the oversight. We have now corrected this, many thanks.

(c) The authors should refrain from using colloquial language such as: “All needed ingredients are on the table, and one can now cook the cake!”

Answer: We fully agreed with the Referee and have now completely rephrased the corresponding sentence.

(d) In light of the analogue of an error function, it would make sense to use a factor of 1/2 in the formula of E_{λ_i} . Then, the entries in Fig.2 with scale to the range [0,1].

Answer: We fully agree with the Referee that the factor $\frac{1}{2}$ would make the formula of E_{λ_i} more elegant.

However, since this is more aesthetic than substantial, and that applying this change force us to make several modifications to the text and redraw several Figures, we have eventually decided to stay with the original formulation. We hope this is acceptable.

We thank the Referee, in any case, for this comment. We will, indeed, certainly use the Referee’s suggestion in all our future submissions (our Manuscript describes completely the transition to synchronization in networks, and at the same time opens up a series of questions that we are considering for future work).

(e) The wording is awkward in parts:

- i. order parameter -> control parameter
- ii. theorematically
- iii. demonstrate how high is

- iv. One has that L is a symmetric, zero row sum, matrix. As so, it is diagonalizable...
- v. our Supplementary Information text,
- v. both calculated along the synchronization solution's trajectory.
- vi. p.4 orthogonal \rightarrow orthonormal
- vii. ij entry vs. (ij) entry

Answer: We hope to have improved the text readability and consistency, and we fixed the wording on this list.

(f) The full account of the considered systems should be given in terms of the parameters.

Answer: This is now done in the revised Manuscript.

Referee: To sum up, I can see that the presented approach to synchronization might have its merits and might warrant publication in Nature Communications. For this, however, the study needs to gain in size by considering more general networks and models of the local dynamics.

Answer: We thank this Referee very much for their very positive evaluation of our work, as well as for the many insightful comments they gave us, that guided the revision of our Manuscript, and allowed us to improve our results, their generality, and the clarity of their presentation.

On the basis of the above answers and the revisions made in the paper (prompted by the Referee's comments and suggestions), we hope that this Referee will now recommend accepting our Manuscript for Nature Communications.

----- 0 -----

REVIEWER COMMENTS

Reviewer #1 (Remarks to the Author):

The authors have considered the initial comments of the referees and made significant revisions to define a more accurate theoretical framework. This revision has improved the work, and I appreciate the efforts of the authors.

However, the authors still claim that “the transition to synchronization of a generic networked dynamical system is a feature that only depends on the topology of the network connections”. This is false, in general. The authors seem to neglect that the critical point ν^* used in their method (summarized from line 395) depends on the node dynamics. In other words, from a theoretical standpoint, the analysis they carry out with their method is totally equivalent to the analysis one can perform by using standard MSF-based methods and symmetries. From a computational standpoint, there can be some differences, but this aspect is completely neglected in the paper. Are there any computational advantages in using the S_n matrices instead of the matrices used in the works by Sorrentino et al or by Motter et al?

As mentioned in my previous report, the novelty and importance of this work are somewhat overstated. The methods are interesting, but incremental with respect to previous works of the same authors and others. As a scientist working on network models, I certainly appreciate the value of simplified models and methodology developments to explore universal laws. However, there is tendency to overstate their power and under-estimate the importance of other factors. It could be even misleading to over-extrapolate from such simplified models to more realistic situations: there are many works in network science that use real-world systems as motivation, but rarely loop back to solve real-world problems to show the values of the modeling and analysis. In Nature Communications I would expect to see examples of applications of a given method to real systems, of broad interest. The authors propose examples that are not realistic. There is no ground truth to be used as a reference and to check if the proposed results are meaningful or not. Using a generic dynamical system to set the node dynamics of a power grid, a biological network or a social network does not provide a “real-world network”, as instead claimed by the authors.

About undirected networks, the authors in their reply say that the MSF approach was extended to asymmetric networks. This is true, but their method assumes that the eigenvalues of the Laplacian matrix are real. What if this assumption is not satisfied? While Laplacians of undirected graphs are diagonalizable and have real eigenvalues, neither statement is necessarily true for Laplacians of digraphs.

Reviewer #2 (Remarks to the Author):

I think the authors have significantly improved the papers along the suggested lines. In particular the existence of symmetries in real-world networks as well as the extended generality of the Laplacian type of coupling in the dynamics are much better justified now.

Thus, I'm ready to recommend the paper for publication in Nat. Comm.

Reviewer #4 (Remarks to the Author):

Report on revised manuscript NCOMMS-23-11343A The transition to synchronization of networked systems by A. Bayani et al.

In their revised manuscript, the authors included several changes to the presentation and extended their approach to two additional applications (a biological and a social network). The changes concerned all parts of the manuscript, from the abstract to the conclusions and supplementary information. For the main text, this resulted in almost two additional pages, including an extended Fig.3 and a new Fig.6. The reference list has been reworked and a number of new references have been added.

In particular, I appreciate the additional panels in Fig.3, which nicely supplement the scan of parameter d . Similarly, I find the revisions to the text helpful as they clarify the study in several aspects.

Concerning the comments in my previous report:

Comment 1: The authors adequately responded to my question, but have not included their reasoning in the manuscript. They indicated the intention to publish related results elsewhere, which is fine in principle, but a short outlook or discussion should be added to the present manuscript.

Comment 2: Again, the authors responded in an adequate manner and extended the text below Eq.(1) for clarification. In that new text passage, the notation should be reworked: At the moment, the function g has the same argument twice. Instead, aiming for generality, I suggest using $g(\{x_1, \dots, x_N\})$.

Comment 3: The authors addressed my suggestion in an elegant way and included additional panels in Fig. 3

Comment 4: I can now see the reason for focusing on chaotic system, and believe that it would add to clarity for the readership if the authors' argument was added to the text, possibly at the point where the systems are introduced (above Eq.(3)).

Comment 5: I agree with the answers that my comment exceeds the scope of the present study and encourage the authors to explore the matter in a future project.

Comment 6: The authors responded by adding new material to the Supplementary Information (new section, explicit adjacency matrix, an additional toy example...). This certainly helps. My point, however, was a different one: I find the start of the section "SYNTHETIC NETWORKS OF LARGE SIZE" too short to understand (or at least get an idea for) the construction of the considered network. The readers would have to dig deep into Ref. [38]. Then, they would learn that the clusters are formed by node of the same degree. I suggest to extend the paragraph by that important piece of information and the essence of the construction scheme.

Comment 7: The authors responded by adding results of two more real-world network, one from biology and one from a social ego-Facebook network. This supports the general relevance of their findings.

Comment 8: In response to my comment, the authors extended the abstract and conclusions by additional sentences stressing the focus on eigenvalues and eigenvectors of the graph's Laplacian matrix. The authors should not devalue the importance of the local node dynamics as it still contributes to the question of synchrony by means of the Jacobian matrices.

All minor issues have been solved.

Finally, one more point that I noticed upon reading the revised manuscript: The author state that the Figs. 3-5 (and presumably Fig.6 as well) depicted "ensemble averages" over a number of different numerical simulations. This triggers the question of standard deviations and an observed error range E_{cl} . Maybe, those could be added as shaded areas in the plots.

Overall, I can see that the manuscript has been further improved and the new results enhance the width of the study. Still, there are a few points that should not only be left in the communication with the Referee(s), but would be worth sharing with the readership who might have similar thoughts. Once these points will have been added in a minor revision, I believe that the manuscript will be fit for publication in Nature Communications.

REVIEWER COMMENTS

Reviewer #1 (Remarks to the Author)

Referee: The authors have considered the initial comments of the referees and made significant revisions to define a more accurate theoretical framework. This revision has improved the work, and I appreciate the efforts of the authors.

Answer: We are very pleased to learn that the Referee considers the revised Manuscript a significant improvement of our work, and appreciates the efforts made by us to fulfill the initial comments of all reviewers. In the following, we address the remaining concerns of this Referee, and hope they will agree with the other two reviewers about the suitability of our Manuscript for publication in Nature Communications.

Referee: However, the authors still claim that “the transition to synchronization of a generic networked dynamical system is a feature that only depends on the topology of the network connections”. This is false, in general. The authors seem to neglect that the critical point v_u^* used in their method (summarized from line 395) depends on the node dynamics.

Response: It is true that the node dynamics (together with the output function) determines the synchronization class, and also the specific value of v^* (for a Class II system). However, once we assume that the system is in Class II, **all** the observed clusters and **their order in the transition** are completely independent on the node dynamics. This is, for instance, seen in Figs. 3a and 3b (which, indeed, were obtained from two different dynamical systems, namely Lorenz and Rössler). The two panels are qualitative equivalent, modulo a rescaling of the horizontal axis. This will be the case for any other (Class II) dynamical system: only the specific value of v^* i.e., the rescaling factor for the horizontal axis, will change, but the cluster synchronization qualitative properties (namely, observed clusters and their order in the transition), will be the same. Therefore, we stand by our statement “*the transition to synchronization of a generic networked dynamical system is a feature that only depends on the topology of the network connections*”, in the sense explained above. Nevertheless, we have decided to further stress this point, and added a new sentence in the Conclusions section.

Referee: In other words, from a theoretical standpoint, the analysis they carry out with their method is totally equivalent to the analysis one can perform by using standard MSF-based methods and symmetries. From a computational standpoint, there can be some differences, but this aspect is completely neglected in the paper. Are there any computational advantages in using the S_n matrices instead of the matrices used in the works by Sorrentino et al or by Motter et al?

Response: We respectfully, but strongly, disagree, as, in fact, our analysis is conceptually different. We tried to address this in our previous response, and significantly expanded our manuscript, in particular the Supplementary Information (SI), namely the new Section 2.2, including Figures 1 to 3. Let us nevertheless reiterate here the conceptual differences with previous work and, unless we are missing something, we hope this reassures the reviewer.

Standard MSF-based analysis identifies clusters, in the sense of subsets of vertices which, under the right conditions, can support synchronization, using symmetries (the automorphism group of the graph, and its subgroups). It **does not** give the **order** in which these clusters will appear as the coupling strength increases from zero, under arbitrary initial conditions. Moreover, the clusters are detected using symmetries, missing on **equitable clusters**, which can also support synchronization

but do not correspond to symmetries. All in all, the standard MSF analysis based on symmetries (a) misses equitable clusters not corresponding to symmetric orbits (orbits of the automorphism group); (b) does not guarantee that some of those identified clusters will actually be realized (as the coupling strength increases from zero, under arbitrary initial conditions). We tried to illustrate this in Figure 1 in the SI, reproduced here for convenience

Figure 1: Schematic Venn diagram showing the relation between the concepts of Equitable Cluster (EC), Symmetric Orbit (SO), and Strong Equitable Cluster (SEC). Note that SOs are equitable, but not necessary strongly equitable (we call those that are, such as those in the statement of Theorem 2.3, Strongly Symmetric Orbits or SSOs). SSOs correspond to symmetric motifs [1, 4] with one orbit. Only SECs (shaded blue) support cluster synchronization in all our examples. Indeed, SECs are guarantee to achieve cluster synchronization when their critical value is reached (see main text). In addition, a SO or EC that is not a SEC on its own, but it is a SEC relative to some clusters, may support synchronization if (all vertices on) those clusters synchronise among themselves; see Fig. 3 for an example.

Note that, the method in Sorrentino et al detect SOs, and more general methods detect (more general) ECs. However, not all of SOs are SECs, which is the name we give to “realisable clusters” or “transient clusters”, that is, group of nodes that actually synchronise as the coupling parameter increases, and not all ECs are SECs either. For example, consider the example shown in Figure 3 (SI), reproduced below.

Figure 3: (Left) A toy network with 33 nodes and 39 links, reproduced from [4], showing small examples of typical

The synchronization order, shown on the right, coincides with that predicted by our Sn-algorithm. The standard MSF analysis based on symmetries would identify the orbits, shown here by colour, but not which of those are realizable, nor in which order. For example, some subsets of orbits are realizable ($\{30,31\}$ or $\{32,33\}$, for example) but not all subsets (no proper subset of $\{11,12,13\}$ or $\{17,18,19,20\}$). We call them relative SECs: a cluster which synchronise relative to another cluster, that is, only after (or at the same time as) another cluster. In Figure 3 above, the red orbit is a SEC relative to the dark blue orbit (and vice-versa), the purple orbit relative to the green orbit (and vice-

versa), and the yellow orbit relative to the pale blue orbit (and vice-versa). Depending on their respective eigenvalues, they may synchronize before, after, or at the same time that the cluster they are relative to. From the SI, again, ‘All in all, being a SEC guarantees cluster synchronization, independently of the rest of the network, when the coupling parameter is high enough (indeed, when its critical value is reached), while being a SEC relative to clusters C_1, \dots, C_m guarantees cluster synchronization when C_1 to C_m are synchronized and the coupling parameter is high enough. (Note that, for simplicity, this discussion only considers Type II systems, see Fig. 1 in the main text.)’ and ‘We can now clarify the relation between SEC and (external) equitable partition. If C_1, \dots, C_r is an equitable partition with r cells, each cell C_i is a SEC relative to the other cells. This guarantees synchronization when the critical values of all the cells are reached, but not necessarily before. In particular, it doesn’t explain the order at which the cluster synchronization occurs, nor partial synchronization of, nor merger between, subsets of the equitable cells, as detected by the S_n algorithm presented in the main text.’ And finally ‘In the theoretical and numerical analysis of the present article, all the cluster synchronization examples found correspond to SECs, or in SECs relative to synchronized clusters, confirming our intuition (see Figs. 2, 3). We conjecture that cluster synchronization can only occur in those cases.’

Of course, we can engineer cluster synchronization by choosing appropriate initial conditions (that is, indeed, the point in Sorrentino et al, that those clusters can support synchronization, under the right conditions). Our S_n -method (a) detects synchronized clusters whether they are symmetry-induced, or not, since they correspond to eigenvector localization, which is more general than symmetry-induced eigenvector localization; and (b) detects synchronized clusters that are realizable (not all symmetry orbit, nor equitable clusters, are, as per discussion above).

In particular, note that the method of Sorrentino (Motter) tries to assess the stability of the cluster solutions that corresponds to a-priori known symmetry orbits (or equitable partitions). In our case, instead, no a-priori knowledge is needed, and the sequence of the clusters appearing in the transition is directly inferred by the inspection of the S_n matrices. In fact, judging by the discussion above, reproduced in the SI (Section 2.2), the very fact of determining a-priori clusters which will synchronize (let alone the synchronization order) is more subtle for symmetric motifs with more than one orbit, which are common in real-world complex networks.

Referee: As mentioned in my previous report, the novelty and importance of this work are somewhat overstated. The methods are interesting, but incremental with respect to previous works of the same authors and others. As a scientist working on network models, I certainly appreciate the value of simplified models and methodology developments to explore universal laws. However, there is tendency to overstate their power and under-estimate the importance of other factors. It could be even misleading to over-extrapolate from such simplified models to more realistic situations: there are many works in network science that use real-world systems as motivation, but rarely loop back to solve real-world problems to show the values of the modeling and analysis. In Nature Communications I would expect to see examples of applications of a given method to real systems, of broad interest. The authors propose examples that are not realistic. There is no ground truth to be used as a reference and to check if the proposed results are meaningful or not. Using a generic dynamical system to set the node dynamics of a power grid, a biological network or a social network does not provide a “real-world network”, as instead claimed by the authors.

Response: We hope our previous response covers this point. Just to reiterate, the main novel message of our work consists of two statements:

- a) the transition to synchronization, in Class II systems, that is, the sequence of synchronized clusters in the order they occur as the coupling strength increases, is **independent** on the specific dynamical system ruling the evolution of the network's nodes, and only depends, instead, **on the spectrum of the network's Laplacian matrix**. This, in particular, applies to real-world networks so we can make predictions on what the transition would be on those networks (which cluster would appear, in which sequence, etc.) without having to worry about whether or not the considered model equations are accurately describing the dynamics of each network's node;
- b) the events that are characterizing the transition to synchronization involve clusters of nodes that **"receive an equal dynamical input"** from the rest of the system. This latter fact can be formalized (see the SI) and constitutes a change of paradigm in the study of cluster synchronization (not orbits, not equitable partitions), and, as seen in the proof of the Main Theorem, is rooted into a localization of a group of Laplacian's eigenvectors.

In terms of applications to real-world networks, note that the toy example Figure 3 represents the type of symmetries found in real-world networks (see e.g. Refs [23-25] in the Main Text), in particular symmetric motifs with more than one orbit, where point (b) above (also, compare to the previous response) is more relevant.

Therefore, for real-world networks **in particular**, the Sn-algorithm may be able to detect cluster synchronization hidden from the standard symmetry approach.

Referee: About undirected networks, the authors in their reply say that the MSF approach was extended to asymmetric networks. This is true, but their method assumes that the eigenvalues of the Laplacian matrix are real. What if this assumption is not satisfied? While Laplacians of undirected graphs are diagonalizable and have real eigenvalues, neither statement is necessarily true for Laplacians of digraphs.

Response: Indeed, the MSF approach has been extended to non-diagonalizable matrices in Ref. [T. Nishikawa and A.E. Motter, Synchronization is optimal in non-diagonalizable networks, Phys. Rev. E **73**, 065106 (2006), Rapid Communications], where a Jordan representation is shown to be sufficient to guarantee stability to the complete synchronization solution (thus extending the MSF to, in principle, any network).

In terms of the stability of cluster synchronization (i.e., of a solution limited to a subset of the network's nodes being synchronized) the critical issue is not having real eigenvalues but rather to have an orthonormal eigen basis, which is not guaranteed for directed, or asymmetric, networks where, even when the diagonalization condition is met in the complex realm, eigenvectors are linearly independent but not necessarily orthonormal. If there is no orthonormal eigen basis, the equations for the components of the synchronization error on each eigenmode are not "variational", and different eigenmodes can be coupled to each other, so that one cannot (in general) reduce such equations into a single parametric one (which is the main point behind the MSF approach). That is why, in these cases, one turns to the Jordan representation.

Now, the situation that is described above has an easy handling for the complete synchronization solution (the one described in the aforementioned reference), but it brings a lot of problems at the

moment of assessing stability of partial, clustered, states. We would like to reassure the Referee that we are aware of these limitations but feel that addressing them go far beyond the scope of the present Manuscript.

We would like to thank this Referee for the stimulating discussion, and the improvements on our manuscript that it prompted. We hope that, in the light of our answers above, we have managed to reassure this reviewer about any remaining issues.

----- O -----

Reviewer #2 (Remarks to the Author)

Referee: I think the authors have significantly improved the papers along the suggested lines. In particular the existence of symmetries in real-world networks as well as the extended generality of the Laplacian type of coupling in the dynamics are much better justified now. Thus, I'm ready to recommend the paper for publication in Nat. Comm.

Response: We thank the Referee for the time and effort spent in reviewing our Manuscript. We are very pleased to learn that the Referee is now fully supporting publication of our Manuscript in Nature Communications.

----- 0 -----

Reviewer #4 (Remarks to the Author):

Report on revised manuscript NCOMMS-23-11343A The transition to synchronization of networked systems by A. Bayani et al.

Referee: In their revised manuscript, the authors included several changes to the presentation and extended their approach to two additional applications (a biological and a social network). The changes concerned all parts of the manuscript, from the abstract to the conclusions and supplementary information. For the main text, this resulted in almost two additional pages, including an extended Fig.3 and a new Fig.6. The reference list has been reworked and a number of new reference have been added.

In particular, I appreciate the additional panels in Fig.3, which nicely supplement the scan of parameter d . Similarly, I find the revisions to the text helpful as they clarify the study in several aspects.

Response: We thank this Referee for his very positive assessment on our revised Manuscript. We are very pleased to see that the Referee fully appreciated our efforts to make the presentation clearer and of higher quality. We also are grateful for the various other points raised by the Referee, and we have made further additions to the main text to address all the Referee's suggestions. Below, we provide a point-to-point answer, including details of the changes made.

Referee: The authors adequately responded to my question but have not included their reasoning in the manuscript. They indicated the intention to publish related results elsewhere, which is fine in principle, but a short outlook or discussion should be added to the present manuscript.

Response: A brief account of the case of multiple regions of stability is now included in the text, just after the descriptions of the three classes of the Master Stability Function.

Referee: Again, the authors responded in an adequate manner and extended the text below Eq.(1) for clarification. In that new text passage, the notation should be reworked: At the moment, the function g has the same argument twice. Instead, aiming for generality, I suggest using $g(\{x_1, \dots, x_N\})$.

Response: The Referee is absolutely right, and we thank them for having pointed out this typo. The notation has now been properly reworked.

Referee: The authors addressed my suggestion in an elegant way and included additional panels in Fig. 3.

Response: We are happy to see that the Referee considers that we have properly fulfilled their important remark.

Referee: I can now see the reason for focusing on chaotic system, and believe that it would add to clarity for the readership if the authors' argument was added to the text, possibly at the point where the systems are introduced (above Eq.(3)).

Response: A discussion on the problems arising when considering periodic dynamics is now added to the text, as suggested correctly by the Referee, just above Eq. (3).

Referee: I agree with the answers that my comment exceeds the scope of the present study and encourage the authors to explore the matter in a future project.

Response: We thank again the Referee for the comment made. We would like to ensure them that we have taken their comment into serious consideration, and we will certainly explore the bubbling phenomenon in a future project.

Referee: The authors responded by adding new material to the Supplementary Information (new section, explicit adjacency matrix, an additional toy example...). This certainly helps. My point, however, was a different one: I find the start of the section “SYNTHETIC NETWORKS OF LARGE SIZE” too short to understand (or at least get an idea for) the construction of the considered network. The readers would have to dig deep into Ref. [38]. Then, they would learn that the clusters are formed by node of the same degree. I suggest to extend the paragraph by that important piece of information and the essence of the construction scheme.

Response: As suggested by the Referee, the section “SYNTHETIC NETWORKS OF LARGE SIZE” has been extended, giving the essence of the construction schemes, and specifying explicitly the considered clusters are formed by nodes of the same degree.

Referee: The authors responded by adding results of two more real-world network, one from biology and one from a social ego-Facebook network. This supports the general relevance of their findings.

Response: We are happy to see that the Referee considers that we have properly fulfilled their important comment.

Referee: In response to my comment, the authors extended the abstract and conclusions by additional sentences stressing the focus on eigenvalues and eigenvectors of the graph’s Laplacian matrix. The authors should not devalue the importance of the local node dynamics as it still contributes to the question of synchrony by means of the Jacobian matrices.

Response: We fully agree with the Referee that the importance of the local node dynamics (and of the coupling function) should not be diminished, as it determines the class of synchronization in the Master Stability Function and the specific critical values of the parameter ν .

We have therefore properly rephrased the sentence of the conclusive part, as requested by the Referee.

Referee: All minor issues have been solved.

Finally, one more point that I noticed upon reading the revised manuscript: The author state that the Figs. 3-5 (and presumably Fig.6 as well) depicted “ensemble averages” over a number of different numerical simulations. This triggers the question of standard deviations and an observed error range E_{cl} . Maybe, those could be added as shaded areas in the plots.

Response: We thank the Referee for having raised this point. Obviously, we do have the data regarding the error ranges of E_{cl} for Figs. 3-6. The reason why we decided not to report them explicitly in the Figures can be easily explained.

Those errors, indeed, refer to the variability (from one simulation to another) of the measured synchronization error in each cluster. Now, since in some cases the clustered are formed by a very limited number of nodes, this variability can be actually rather large during the unsynchronized regime, and reporting it as a shaded area would have the effect of making the Figure harder to interpret. Nevertheless, what is really crucial to point out is that the specific value of E_{cl} has no importance in the unsynchronized regime: the only thing that matters is that E_{cl} is strictly different than zero.

On the other hand, what we observe is that the variability (and the associated error range) vanishes exactly when E_{cl} vanishes i.e., **in all simulations the synchronous regime is attained for the same value of the coupling strength**. This allows us to define in a very robust way the synchronization condition ($E_{cl} = 0$).

To illustrate what we just stated, we report here below the case of the US-power grid (800 simulations), and the Referee can see that the errors (the shaded areas) go exactly to zero when E_{cl} goes to zero, whereas they are rather large during the unsynchronized stages of the dynamics.

USA power grid network ($N = 4941$), $n_{sims} = 800$

Moreover, looking at the figure, one can see that reporting the variability as a shaded area could obscure the plot making it harder to clearly distinguish the synchrony order of the clusters.

Having said this, we agree with the Referee that the fact that the variability vanishes when the error vanishes is indeed a crucial information not to be hidden in our presentation, and consequently we have added this information to the main text.

Overall, I can see that the manuscript has been further improved and the new results enhance the width of the study. Still, there are a few points that should not only be left in the communication with the Referee(s) but would be worth sharing with the readership who might have similar thoughts. Once these points will have been added in a minor revision, I believe that the manuscript will be fit for publication in Nature Communications.

Response: We believe that we have implemented all needed minor changes in the text to share with the readership the improvements in the manuscript following the reviewing process. We are confident to have properly addressed all the Referee's recommendations, and we are happy to learn that the Referee is supporting publication of our Manuscript in Nature Communications.

----- o -----

REVIEWER COMMENTS

Reviewer #1 (Remarks to the Author):

The authors tried to “reassure” me, but I still see a fundamental bug in their method.

It is on page 3, lines 191-221. These sentences are correct for global synchronization, but in the case of cluster synchronization, they are incorrect, in general.

In Reference [17] (Pecora, L. M., Sorrentino, F., Hagerstrom, A. M., Murphy, T. E., & Roy, R. (2014). Cluster synchronization and isolated desynchronization in complex networks with symmetries. *Nature Communications*, 5(1), 4079.) and following papers it is NOT claimed that the cluster stability depends on certain eigenvalues/eigenvectors of the Laplacian matrix. By contrast, they say that:

- By changing the clusters, the quotient network changes, and therefore also the set of parallel and transverse perturbations. The change of the quotient network changes the synchronous solution about which the dynamics are linearized. This means that, in the presence of synchronous clusters, there are more synchronous solutions around which applying linearization.
- By increasing the number of clusters, the number of transverse perturbations that determine the stability of the synchronous solution decreases, i.e., the dimension of the transverse space changes with the number of clusters.
- The presence of multiple synchronous solutions (one per cluster) makes the decomposition of the perturbations in “independent modes” more difficult, and therefore diagonalizing the Laplacian matrix is not possible/sufficient. To solve this problem, many techniques have been proposed by Sorrentino et al and by Motter et al. In any of these papers, it is stated that it is sufficient to look at some eigenvalues/eigenvectors of the Laplacian matrix to study the stability of the cluster synchronous solutions, as claimed by the authors in the cited lines.

In summary, the cluster-synchronous solution depends on the quotient network, and therefore on topology, node dynamics, and clusterization. This means that the value ν^* found by the authors is not the same for global synchronization and cluster synchronization, in general. Moreover, it changes for any clusterization. Therefore, by changing clusterization, we can have a different ν^* , but also a different MSF shape, and even a different class. This implies that also the order of the transitions can change. This also means that their results are wrong, in general. The absence of comparisons with a ground-truth example (as required in my previous review) does not play in favor of the authors. In the very particular case in which ν^* is the same for global synchronization and cluster synchronization, the results provided by their method are the same as those provided by methods already available.

As an example, you can consider a network with 100 nodes, with in-degree 100, and Roessler oscillators with $a = 0.1$, $b = 0.1$, $c = 5$.

You can check that both the MSF (and even its class!) and the ν^* value depend on the considered clusterization (see attached figure).

Moreover, the sentence “Our work exhibits a higher level of generality and efficiency with respect to other algorithms assessing the stability of network’s symmetry orbits [17, 20], as our method does not require an a-priori knowledge of the network’s symmetries.” is wrong. The cited methods and their evolutions, such as

-) Cho, Y. S., Nishikawa, T. & Motter, A. E. Stable chimeras and independently synchronizable clusters. *Phys. Rev. Lett.* 119, 084101 (2017).

-) Della Rossa, F., Pecora, L., Blaha, K., Shirin, A., Klickstein, I., & Sorrentino, F. Symmetries and cluster synchronization in multilayer networks. *Nature Communications*, 11(1), 3179 (2020).

-) Zhang, Y. & Motter, A. E. Symmetry-independent stability analysis of synchronization patterns. *SIAM Rev.* 62, 817–836 (2020).

-) Lodi, M., Sorrentino, F., & Storace, M. One-way dependent clusters and stability of cluster synchronization in directed networks. *Nature Communications*, 12(1), 4073 (2021).

are more general, as they allow one to study the stability of synchronous cluster solutions even when the quotient network changes. Moreover, if we assume that ν^* does not change with the clusterization, these methods provide the same results proposed by the authors. For example, let us consider the network studied by the Authors in Fig. 2. We assume that the network has a Master Stability Function of type 2, which intersects the ν axis at ν^* (as assumed in the paper). In this particular case, the changes of ν^* are small, therefore the proposed results can be compared with those obtained with other methods, under the (wrong!) assumption that ν^* does not change.

If one applies the method described in Reference [17] starting from global synchrony, the same results proposed by the authors are obtained (I checked it). There is no novelty.

Therefore, I still claim that the opening sentence of the abstract “We show that the transition to synchronization of a generic networked dynamical system is a feature that only depends on the topology of the network's connections, and can be entirely predicted and completely characterized with the only help of eigenvalues and eigenvectors of the graph's Laplacian matrix.” is false. This is my main point, and I am not reassured at all by the authors’ reply.

Secondarily, I still claim that the authors propose examples that are not realistic. There is no ground truth to be used as a reference and to check if the proposed results are meaningful or not. Using a generic dynamical system to set the node dynamics of a power grid, a biological network or a social network does not provide a “real-world network”, as instead claimed by the authors.

Finally, in some specific cases, the Laplacian properties may be sufficient to analyze the stability of synchronous solutions, but the Authors should clearly state under which assumptions about the topology and dynamics of the considered network. Otherwise, their paper does not add anything to the state of the art.

Reviewer #4 (Remarks to the Author):

Report on Nature Communications manuscript NCOMMS-23-11343B and the report of Reviewer 1:

After reading the revised manuscript again and going through the reasoning of Reviewer 1, here are my comments:

1. I side with Reviewer 1 in terms of their comment on transferring the MSF arguments from complete synchronization to cluster synchronization. In the latter, the MSF changes and needs to be recomputed for the specific synchronized dynamics under consideration.

a. Therefore, lines 212 to 221 are misinterpretations of the findings of Ref.[17] and should be deleted.

b. In other words, If those lines are removed, the section THE SYNCHRONIZATION SOLUTION are only about complete synchronization. The transformation from Eq.(1) to Eq.(3) in the Supplementary material requires more thought and care in the case of cluster synchronization.

c. As a consequence, the first sentence in the abstract is omissive and too general. It neglects that the decomposition of perturbations changes and so does the linearization around the cluster state at hand.

d. As an effect, the eigenvectors of the original Laplacian still form a basis, but a subset of them does not necessarily span the tangent space of the (cluster) synchronous solution as the text around Eq.(2) might suggest.

2. I also agree with Reviewer 1 that the value ν^* can change for different (cluster) synchronized solutions, if that values is interpreted as the critical value of the stability-instability transition of the cluster solution. The case, when it stays fixed, is the exception rather than the rule.

3. Speaking of Ref.[17], the authors of that paper work with a more general framework, which goes beyond Laplacian coupling matrices.

4. Finally, I disagree with Reviewer 1 that ground truth example needs to be based on an experiment. Numerical simulations for a network extracted from real-world data would do. Having said this, a more complex example than a 10-node, all-to-all network is needed as presented by the authors. Still I have the concern, that a chaotic dynamical model applied to arbitrary real-world networks has little relevance.

To move forward, I suggest that the authors carefully revised the framing and expressions with the goal of more mathematical rigor.

i) This applies to Section “THE PATH TO SYNCHRONIZATION” in particular. Then, the unfolding of tangent spaces in relation to the respective coupling and block-diagonalized matrices could be made rigorous and complete.

ii) Subsequently, statements like

“Laplacian matrix L uniquely defines G , and as so any clustering property of the network G has to be reflected into a corresponding spectral feature of L ”

could be phrased with more care and less room for misinterpretations.

iii) Similarly, the authors should clarify the meaning of ν^* . My understanding is that it refers to complete synchrony only. Along the same lines, the notion of a “cluster” could be stated more clearly, i.e., as a definition.

iv) Moreover, the authors should clearly state, what kind of coupling (diffusive?) their framework is restricted to.

v) Finally, it might be helpful if the authors consider the network examples of Ref.[17] as additional cases.

To sum up, I rather see an issue with the presentation of the modelling framework than the soundness of the study. The presented transition towards complete synchronization is still worth reporting, but the

theoretical framework might not be as general as the authors claim in their revision. Proper credit should be given to previous works to relate to the current study.

RESPONSE TO REVIEWERS' COMMENTS

Reviewers #1 and #4

First, we would like to thank the two reviewers for the comments: as it will appear momentarily, they have prompted a rather large revision of our Manuscript, and they have contributed (in our opinion) to improve substantially the clarity, quality, and rigor of our presentation.

That is the reason why we felt the need to explicitly acknowledge (in the main text) the role played by the two Referees in guiding the writing of what we hope to be the final version of our Manuscript.

Second, we have carefully considered the criticism made by Referee #1, and we must admit that, rigorously speaking, this Referee is right: the Master Stability Function (MSF) formalism is only valid for the complete synchronization scenario, where all the nodes in the network adjust their trajectory into the one of a single, isolated system.

In contrast, when one considers cluster-synchronous states within a network, the trajectory followed by the clustered nodes depends explicitly on the rest of the network. Hence, the assessment by Referee #1, echoed by Referee #4, that the MSF arguments for global synchronization do not immediately apply to cluster synchronization, is correct.

We have, therefore, thoroughly revised our manuscript in order to

1. Explain rigorously the stability properties of the cluster solutions in the context of the MSF formalism; and
2. Modify the article to clarify the scope and generality of our theoretical framework.

We decided, therefore, to start our revision by adding a new subsection in the Supplementary Material, which reviews the stability properties of the different cluster solutions considered in the Manuscript, and clarifies the difference, in particular the fact that the use of the very same MSF formalism doesn't make sense in principle (i.e., rigorously) here, as it would not be possible to formally separate dynamics and structure.

It does, however, provide a useful approximation of the critical coupling strengths, without affecting the other claims in our manuscript, as we explain below.

For clarity in our remaining answers, we report here, below, the text of the subsection that we have added (note that the notation used is the same as in the paper).

The stability properties of the clusters' synchronous solution

It is important to remark that all the above results are formally valid only for the whole network's synchronous solution. The trajectories followed by the nodes' dynamics in each cluster-synchronous state slightly differ, instead, from those which are followed in the global solution, as they rigorously

depend on the quotient network, and therefore on topology, node dynamics, and clusterization. Let us indeed focus on a given cluster C_1 , and let us call C_1 the set of indices identifying the nodes that belong to C_1 . For each node i ($i \in C_1$), the equation of motion is

$$\dot{\mathbf{x}}_i = \mathbf{f}(\mathbf{x}_i) - d \sum_{j \in C_1} \mathcal{L}_{ij} \mathbf{g}(\mathbf{x}_j) - d \sum_{j \notin C_1} \mathcal{L}_{ij} \mathbf{g}(\mathbf{x}_j), \quad (1)$$

where the overall coupling is now split into the sum of an intra-cluster term and of a term accounting for the connections of the cluster to the rest of the network.

Eq. (1) can be rewritten as

$$\dot{\mathbf{x}}_i = \mathbf{f}(\mathbf{x}_i) - d \sum_{j \in C_1} \mathcal{L}_{ij} \mathbf{g}(\mathbf{x}_j) + d \sum_{j \notin C_1} \mathcal{A}_{ij} \mathbf{g}(\mathbf{x}_j), \quad (2)$$

where A is the adjacency matrix. This is because all the elements of the Laplacian matrix considered in the second coupling term are just the opposite of the corresponding terms of the adjacency matrix. The second coupling term is, indeed, limited to $j \notin C_1$ and therefore, by definition, it does not contain the diagonal element of the Laplacian ($j = i$) which is instead contained in the intra-cluster coupling term.

We now recall that the main theorem of our study asserts that synchronizable clusters are those formed by nodes which are equally connected to, and receive an equal input from, the rest of the network. Therefore, as the last term of Eq. (2) accounts for the total input received by node i from all nodes that do not belong to the cluster, our theorem ensures that it is a term which is formally independent on i . The cluster synchronous solution is $x_i(t) = x_j(t) = x_{C_1}(t)$, $\forall i \in C_1$ and $\forall j \in C_1$ ($j \neq i$), and obeys the equation

$$\dot{\mathbf{x}}_{C_1} = \mathbf{f}(\mathbf{x}_{C_1}) + d \sum_{j \notin C_1} \mathcal{A}_{ij} [\mathbf{g}(\mathbf{x}_j) - \mathbf{g}(\mathbf{x}_{C_1})]. \quad (3)$$

This is because the diagonal terms of the Laplacian matrix are equal to the sum of the number of intra-cluster connections and of the number of extra-cluster connections, the latter ones being now incorporated in the remaining coupling term, which once again does not depend on i . Once again, one can consider perturbations $\delta x_i = x_i - x_{C_1}$ ($\forall i \in C_1$) and perform linear stability analysis of Eq. (2). The result is

$$\delta \dot{\mathbf{X}} = [\mathbb{I} \otimes \mathbf{Jf}(\mathbf{x}_{C_1}) - d \mathcal{L} \otimes \mathbf{Jg}(\mathbf{x}_{C_1})] \delta \mathbf{X}, \quad (4)$$

where $\delta \in \mathbb{R}^{N_1}$ is the global error vector, and N_1 is the number of nodes forming the cluster C_1 .

It is immediately seen that the linearized Eq. (4) is formally identical to that ruling the evolution of the error vector in the case of the whole network's synchronous solution, and therefore the same expansion of the error can be made with the eigenvectors of the Laplacian. The difference, however, is that the evaluation of the maximum Lyapunov exponents requires now to calculate the Jacobians of the functions f and g over the cluster-synchronous solution x_{C_1} , which obeys Eq. (3). In other words, while the MSF formalism calculates the Maximum Lyapunov exponents using the trajectories experienced by the whole network's synchronous solution (a state in which each node of the network repeats the same dynamical behavior of a single, isolated, system), the trajectories experienced by the

cluster synchronous state are perturbed by an extra term $K(t) = d \sum_{i \in A} [g(x_i) - g(x_t)]$, and depend therefore explicitly on the entire network's dynamics, and on the specific coupling function. This fact leads to two main consequences:

- The very same cluster-synchronous solution is not invariant, as $K(t)$ explicitly depends on d . In particular, at each value of the coupling strength one would have a distinct cluster-synchronous state, and therefore the entire framework of the MSF would make no sense in this case as it would not be possible to rigorously separate dynamics and structure;
- The perturbation $K(t)$ may lead the synchronous trajectories to visit areas of the phase space which are instead never visited by a single isolated system, and therefore it may determine slight variations in the calculations of the maximum Lyapunov exponents, and consequently slight variations in the determination of the critical coupling strength value at which the cluster synchronous state becomes stable.

On the other hand, the term $K(t)$ is directly proportional to d , and therefore it has to be expected that such a perturbation will, in fact, be small across the transition to complete synchronization, where d starts from 0 and only slightly increases to values which are normally smaller than 1. In addition, $K(t)$ consists of the sum of terms which are in general uncorrelated, as there are no constraints on the dynamics of the nodes which do not form part of the synchronous cluster. This latter fact would also contribute to determine the smallness of the perturbation.

In our study, we decided therefore to adopt a practical approximation to the problem by assuming that the perturbation $K(t)$ is always negligible and, consequently, that the trajectories visited by the clustered synchronous nodes are always equal to those that characterize complete synchronization (i.e., those of a single, isolated, system). This allows one to refer to a unique MSF (the one constructed in the whole network's synchronous state) for determining the stability properties of all cluster synchronized states.

In other words, the trajectory followed by the cluster-synchronous nodes slightly differs from that followed by all nodes of the network during complete synchronization: the latter is the same of a single, isolated, system, the former is the solution of the equation of an isolated system where an additional (perturbative) term exists which account for the coupling with the network nodes not belonging to the cluster (and therefore whose dynamics are not necessarily synchronous).

The case presents, therefore, similarities to the situation that one encounters in “noise induced synchronization” (NIS, Phys. Rev. E 67, 027201 (2003) and Phys. Rev. E 67, 066220 (2003)), a threshold phenomenon where noisy perturbations applied to a chaotic dynamical system may slightly change stability, in that it may drive the trajectory to visit areas of the phase space which are instead prohibited in the noiseless case, and this way it may determine slight variations in the calculations of the maximum Lyapunov exponent.

Once again, we need to thank Referee #1, because they are correct, and our calculation of the critical coupling strengths at which the different cluster-synchronous solutions become stable is actually an approximation.

Therefore, we have reformulated several sentences in the main text, where this point is now explicitly stated and, as suggested explicitly by Referee #4, we have toned down some claims about the generality of our approach in the calculation of the stability thresholds for cluster synchronization.

At the same time, we would like to point out that the approximation of the critical coupling strengths does not affect the other relevant points made in our study. In particular:

- The method which uses the $*$ matrices to reveal all possible synchronizable clusters is entirely valid and is not affected by the considered approximations. Once again, our method introduces the new concept of dynamical equitability (clusters are the sets of nodes receiving equal input from the rest of the network) and allows us to clarify that synchronizable clusters are more general than symmetry orbits and more specific than equitable partitions;
- The identification of the nodes forming part of each of the clusters is not affected either by the approximation, and therefore all our claims about the fact that clusters are independent on the specific dynamical system used are still valid.

Moreover, all our numerical studies demonstrate that the approximations obtained are, in fact, rather good estimates of the critical coupling strength for each of the considered clusters.

Although, as Referee #1 points out, one can manufacture a counterexample, we argue that our approximation is very accurate whenever (as it always occurs in real-world networks) the size of the clustered state is much smaller than the size of the entire network. Indeed, looking at the Figure provided by Referee #1, one immediately sees that the maximal difference observed (about 15% in the value of the critical coupling strength) occurs when the two clusters have the same size (50 nodes), and the cluster size is exactly a half of the size of the entire network.

In summary, we believe we have fully addressed the issue raised by Referee #1 and echoed by Referee #4.

The substantial changes in the manuscript main text, plus the additional section in the Supplementary Material, add mathematical rigor and puts our results in the correct theoretical framework and scope, hopefully once and for all.

We believe, as also Referee #4 does, that our results on the transition toward global synchronization are worth reporting, and we are hopeful that this is the last revision before it can be deemed acceptable for publication in Nature Communications.

----- o -----

REVIEWERS' COMMENTS

Reviewer #4 (Remarks to the Author):

Report on Nature Communications manuscript NCOMMS-23-11343C

In their revised version, the authors accounted for the criticism in the previous reports and aimed for more mathematical precision. They reworded and extended the main text in several places (abstract, Introduction, second section on the synchronous solution...) and added a new section to the Supplemental Information. The latter section now provides detailed information on how the MSF framework is related to the considered cluster synchronization.

Overall, the changes have made the study more accessible and put it in a better light by highlighting the approximative condition of its approach.

It is my opinion that the revised manuscript can be accepted for publication provided that the following aspect is addressed:

The condition that (lines 116-119, cf. Theorem 2.2 in SI)

“all nodes in a cluster have the same connections (and the same weights) with nodes not belonging to the cluster, and therefore they receive the same dynamical input from the rest of the network”

should be mentioned on the Conclusion section as well and should replace the end of the first sentence in the Conclusions:

“for all possible dynamical systems and all possible network’s architectures.”

This aspect can also be seen as a definition of “cluster”. The authors should keep in mind that cluster synchronization can have an alternative interpretation. I am referring to the dynamical scenario where the nodes group into 2, 3, 4 or more groups that are synchronized internally, but not with other groups. In that tradition, an N cluster would refer to a splay state where every node acts differently from the rest. A clarifying sentence would avoid confusion.

In the same light, transferring the statement of the new SI section on distinct cluster states to the end of the second section (line 302) would provide further clarity, e.g.:

“Note that at each value of the coupling strength one might have a distinct cluster-synchronous state.”

Finally, I suggest explicitly stating the eigenvectors to the eigenvalues of the toy examples in an additional SI section.

With the above changes implemented, I will be happy to recommend acceptance.

Point-to-point answer to Referee #4

Ref: In their revised version, the authors accounted for the criticism in the previous reports and aimed for more mathematical precision. They reworded and extended the main text in several places (abstract, Introduction, second section on the synchronous solution...) and added a new section to the Supplemental Information. The latter section now provides detailed information on how the MSF framework is related to the considered cluster synchronization.

Overall, the changes have made the study more accessible and put it in a better light by highlighting the approximative condition of its approach.

It is my opinion that the revised manuscript can be accepted for publication provided that the following aspect is addressed:

Answer: We are happy to see that the Referee agrees that the revision has improved the quality and clarity of our presentation, and that they consider the revised Manuscript suitable for publication. Moreover, we take this occasion to thank once again the Referee for the additional suggestions contained in their report, that we have gladly implemented in the final version of the Manuscript. Here below, we provide our answers to the report.

Ref: The condition that (lines 116-119, cf. Theorem 2.2 in SI) “all nodes in a cluster have the same connections (and the same weights) with nodes not belonging to the cluster, and therefore they receive the same dynamical input from the rest of the network” should be mentioned on the Conclusion section as well and should replace the end of the first sentence in the Conclusions:

“for all possible dynamical systems and all possible network’s architectures.”

This aspect can also be seen as a definition of “cluster”. The authors should keep in mind that cluster synchronization can have an alternative interpretation. I am referring to the dynamical scenario where the nodes group into 2, 3, 4 or more groups that are synchronized internally, but not with other groups. In that tradition, an N cluster would refer to a splay state where every node acts differently from the rest. A clarifying sentence would avoid confusion.

Answer: We fully agree with the Referee.

We have added the suggested sentence in the first paragraph of the Conclusion section (now headed as the Discussion section due to a specific editorial request) exactly in the way the Referee requested us to do (i.e., by replacing part of the existing text).

Ref: In the same light, transferring the statement of the new SI section on distinct cluster states to the end of the second section (line 302) would provide further clarity, e.g.:

“Note that at each value of the coupling strength one might have a distinct cluster-synchronous state.”

Answer: Once again, we fully agree with the Referee and have added the suggested sentence at the end of the second section.

Ref: Finally, I suggest explicitly stating the eigenvectors to the eigenvalues of the toy examples in an additional SI section.

Answer: The Referee here refers to the toy models depicted in Figs. 2 and 3 of the Supplementary Material.

As for the toy model of Figure 2, we notice that the values of the 8 eigenvalues were already stated in the caption of the Figure, and we guess the Referee may have possibly overlooked them.

As for the toy model of Figure 3, we instead fully agree with the Referee, and have added the values of the 33 eigenvalues to the caption of the Figure.

However, eigenvectors of both systems haven't been included for practical reasons. As they are too long to display (they are, respectively, 8×8 and 33×33 matrices) and are easily computed from the Laplacian matrices if needed.

Ref: With the above changes implemented, I will be happy to recommend acceptance.

Answer: In summary, we are sincerely grateful to this Referee for the many constructive suggestions they have done along the entire review process of our Manuscript.

We, indeed, are well aware of how precious such suggestions have been to guide us in the various revisions made of our Manuscript, up to reaching this final version.